# A Spectral Characterization of Generalization in GCN: Escaping the Curse of Dimensionality

## Abstract

Empirically it is observed that Graph Convolution Networks (GCNs) often generalize better than fully connected neural networks (FCNNs) on graph-structured data. While this observation is often attributed to the ability of GCNs to exploit knowledge about the underlying graph structure, a rigorous theoretical explanation remains limited. In this work, we theoretically prove that one factor for the improved generalization of GCNs arises from the spectral representation of the filters or graph convolutional layers. Specifically, we derive generalization bounds that are independent of the number of parameters and instead scale nearly linearly with the number of graph nodes, offering a compelling explanation for their superior performance in over-parameterized regimes. Furthermore, in the limit of infinite number of nodes, we prove that under certain regularity conditions on the spectrum, GCNs escape the curse of dimensionality and continue to generalize well. We demonstrate our conclusions through numerical experiments.

## 1 Introduction

Graph convolutional neural networks (GCNs) (Defferrard et al., 2016; Kipf & Welling, 2017; He et al., 2022) have emerged as a powerful tool for learning from graph-structured data, enabling applications in various domains such as social networks (Fan et al., 2019), recommender systems (Wu et al., 2022), and protein design (Strokach et al., 2020). The remarkable success of GCNs is largely attributed to their superior performance on unseen data compared to FCNNs, when the data has an underlying graph structure (Dwivedi et al., 2023). However, despite strong theoretical support for the use of GCNs in terms of their expressive power (Zhang et al., 2025), stability (Gama et al., 2020), and transferability (Ruiz et al., 2020), a rigorous theoretical understanding of conditions that allow for an improved generalization capability compared to FCNNs is largely unexplored in the literature.

Several existing works (Liao et al., 2020; Tang & Liu, 2023; Wang et al., 2025a) characterizing the generalization of GCNs are built on the classical statistical learning theory frameworks such as VC dimension (Vapnik, 2013), PAC-Bayes (Shawe-Taylor & Williamson, 1997), and Rademacher complexity (Bartlett & Mendelson, 2002). However, these bounds typically scale with the raw parameter count in the network, often resulting in very loose bounds compared to empirical observations. In the context of FCNNs, this has led more recent generalization theory frameworks to explore bounds based on information-theory, algorithmic stability, the so-called "double-descent" phenomenon, and properties of the training loss landscape (Hochreiter & Schmidhuber, 1997; Schaeffer et al., 2024; Hellström et al., 2025). For GCNs, there are very few attempts to apply these more modern techniques. Shi et al. (2024) recently showed that GCNs exhibit a phenomenon where test error first increases and then decreases with the label ratio, but their analysis is limited to simple linear GCNs with one filter tap for community stochastic block models (cSBMs). There lacks a general framework for analyzing the generalization of GCNs beyond the classical approaches.

GCNs are highly structured models that exploit the underlying graph structure using the graph convolution/message passing/filtering operations that share parameters across nodes, unlike FCNNs. This suggests that the *intrinsic* dimension of the convolutional layers is often significantly lower than more basic notions of model dimension like the raw parameter

count. This structure is often not exploited in the literature, and here we argue that the spectral domain of the convolutional filters provides a natural representation for the analysis of GCN generalization. Specifically, leveraging ideas from classical signal processing, we note that filters or convolutional layers can be effectively represented in terms of their frequency response, which lies in a space whose dimension is the number of nodes in the graph rather than the full parameter space. In practice, we need larger number of parameters to have better expressive power for GCNs. This shift in the perspective from the parameter space to the frequency response will be crucial to obtain tighter generalization bounds.

**Paper contributions.** In this work, we formally exploit the spectral structure of GCNs to derive sharper generalization bounds. Specifically,

**(i).** We present a general framework that provides generalization bounds for GCNs on arbitrary graph structures. Our bounds only require computation of well-known and easy-to-compute complexity measures, such as covering numbers in the spectral domain.

**(ii).** We provide a generalization bound which scales as $\sqrt{Ln_x/N}$, where $L$ represents number of layers, $n_x$ represents number of nodes and $N$ is number of data points. Our bounds are independent of the total number of model parameters. These bounds provide an explanation for the empirical success of GCNs even when the number of parameters is comparable to or larger than that of FCNNs. Moreover, these bounds are significantly tighter than state-of-the-art results.

**(iii).** We extend our analysis to graphons (countably infinite sized-graph). By assuming mild spectral regularity, the bounds scale as $N^{-1/6}$ and are dimension-free, thus effectively escaping the curse of dimensionality.

**(iv).** Finally, we corroborate our theoretical insights with numerical simulations.

The remainder of the paper is organized as follows. In §2, we formulate the learning problem, discuss the limitations of existing generalization bounds, and motivate the need for a new approach. In §3, we present our general generalization bounds for GCNs and then present its application in various regimes. We validate our theoretical findings through numerical experiments in §4, and conclude the paper in §5.

**Notation.** The Hermitian (transpose-conjugate) of a matrix $M$ is denoted by $M^{\mathsf{H}}$. The set $\mathbb{O}(n_x, \mathbb{C})$ denotes all $n_x \times n_x$ unitary (orthonormal) matrices with complex entries. The operator $\mathsf{diag}(\cdot)$ constructs a diagonal matrix with the given vector as its diagonal elements, while $\mathsf{diag}^{\dagger}(\cdot)$ denotes the inverse operation, extracting the diagonal entries into a vector.

The abbreviations "w.p." and "a.s." stand for "with probability" and "almost surely", respectively. A function $f$ is said to be Lipschitz continuous if there exists a constant $L > 0$ such that for all $\mathbf{x}_1, \mathbf{x}_2 \in \mathcal{X}$, we have $\|f(\mathbf{x}_1) - f(\mathbf{x}_2)\| \leq L\|\mathbf{x}_1 - \mathbf{x}_2\|$. Moreover, the function $f$ is said to be Lipschitz smooth if its gradient is Lipschitz continuous.

We write $\beta \leq$ (or $\geq$) $\mathcal{O}(\alpha)$ to mean that there exists a constant $C > 0$ such that $\beta \leq$ (or $\geq$) $C\alpha$ for all $\alpha$ in the domain of interest. The notation $[A]$ denotes the index set $\{1, 2, \ldots, A\}$. The set $\mathbb{R}^{\infty}$ represents the set of all countably infinite sequences of real numbers.

## 2 PROBLEM FORMULATION

Consider a graph $\mathcal{G} = (\mathcal{V}, \mathcal{E}, A)$, where $\mathcal{V} \subseteq [n_x]$ is the vertex set, $\mathcal{E} \subseteq [n_x] \times [n_x]$ is the edge set, and $A \in \mathbb{R}^{n_x \times n_x}$ is the adjacency matrix (where $A_{ij} = \mathbb{1}_{\mathcal{E}}((i,j))$). Suppose $\mathbf{x} \in \mathbb{R}^{n_x}$ is a vector supported on the graph $\mathcal{G}$, also referred to as a graph signal, where each entry $x_i$ corresponds to the scalar feature at the $i$-th node. In this work, we are interested in learning GCN that map the graph signal $\mathbf{x}$ to target $\mathbf{y} \in \mathbb{R}^{n_y}$, where the tuple $(\mathbf{x}, \mathbf{y})$ is a random variable drawn from an unknown joint distribution $\mu$.

**Graph Convolutional Neural Networks (GCNs)** are among the first proposed graph neural network (GNN) architectures (Defferrard et al., 2016; Kipf & Welling, 2017). GCNs are constructed by a sequence of compositions of graph convolutional layers, non-linearities, and pooling operations. The graph convolution (or filtering) operation is the key building block of GCNs that aggregates features (or messages) from neighboring nodes in a linear

fashion. Let $\mathcal{H} \subset \mathbb{R}^{\infty}$ be the set of graph filter coefficients. At the layer $l \in [L]$, the convolutional map $\phi_l : \mathcal{H} \times \mathbb{R}^{d_l} \to \mathbb{R}^{d_l}$ is defined as

$$\phi_l(\mathbf{h}, \mathbf{z}) := \sum_{k \in \mathbb{N}} h_k S_l^k \mathbf{z}, \tag{1}$$

where $S_l \in \mathbb{S}^{d_l}$ is called convolutional operator associated with the graph $\mathcal{G}$ at layer $l$ (e.g., $A$ or any Laplacian operator such as $\mathrm{diag}(A\mathbf{1}) - A$; see Defferrard et al. (2016); Kipf & Welling (2017)) and $n_f$ represents the number of filter taps, i.e., the number of non-zero coefficients in $\mathbf{h}$. At each layer $l \in [L]$, we have non-linear activation $\sigma_l : \mathbb{R}^{d_l} \to \mathbb{R}^{d_l}$, (such as ReLU, tanh, or sigmoid functions). Finally, we have a pooling or resampling operation $P_l : \mathbb{R}^{d_{l-1}} \to \mathbb{R}^{d_l}$ that reduces (or increases) the size of the graph at each layer, gradually making the final prediction compatible with the target vector. The intermediate output at layer $l \in [L]$ and channel $c \in [C_l]$ is denoted by $\mathbf{x}_l^c \in \mathbb{R}^{d_l}$ and is computed as

$$\mathbf{x}_l^c = P_l \left( \sigma_l \left( \sum_{g \in [C_{l-1}]} \phi_l(\mathbf{h}_l^{(c,g)}, \mathbf{x}_{l-1}^g) \right) \right). \tag{2}$$

The map $\mathbf{x}_0^1(= \mathbf{x}) \to \mathbf{x}_L^1(= \hat{\mathbf{y}})$ is called the GCN and is denoted by $\Phi : \mathscr{H} \times \mathbb{R}^{n_x} \to \mathbb{R}^{n_y}$, where the set $\mathscr{H} = \{\mathcal{H}^{C_l \times C_{l-1}}\}_{l \in [L]}$ consists of all the filter coefficients. Convolutional neural networks (CNNs) (Denker et al., 1988) are a special case of GCNs, since CNN convolutions operate on regular grid graphs like images, where pixels are nodes (see §A.1).

**Statistical learning problem.** We consider the learning problem of minimizing the un-regularized risk with loss function $\ell : \mathbb{R}^{n_y} \times \mathbb{R}^{n_y} \to \mathbb{R}$, sample space $\Omega$, and measure $\nu$:

$$\mathcal{R}_\nu(\mathbf{H}) := \int_{\omega \in \Omega} \ell(\mathbf{y}(\omega), \Phi(\mathbf{H}(\omega), \mathbf{x}(\omega))) d\nu(\omega). \tag{3}$$

The goal is to minimize the risk evaluated on the probability measure $\mu$, referred to as the population risk, i.e., $\hat{\mathbf{H}}_N \in \arg\min_{\mathbf{H} \in \mathscr{H}} \mathcal{R}_\mu(\mathbf{H})$. However, in practice, complete access to $\mu$ is unavailable. For tractability (Vapnik, 2013), we relax the optimization problem to the empirical risk minimization by using finite data points $\{(\mathbf{x}_i, \mathbf{y}_i)\}_{i=1}^N$ drawn from the distribution $\mu$ that forms an empirical distribution $\mu_N$ and solve the program:

$$\hat{\mathbf{H}}_N \in \arg\min_{\mathbf{H} \in \mathscr{H}} \left\{ \mathcal{R}_{\mu_N}(\mathbf{H}) = \frac{1}{N} \sum_{i \in [N]} \ell(\mathbf{y}_i, \Phi(\mathbf{H}, \mathbf{x}_i)) \right\}. \tag{ERM}$$

There arises a natural question of whether the learned parameters $\hat{\mathbf{H}}_N$ perform well on unseen data; i.e., $\mathcal{R}_\mu(\hat{\mathbf{H}}_N) \approx \mathcal{R}_{\mu_N}(\hat{\mathbf{H}}_N)$? The discrepancy between the population and empirical risk is called *Generalization Error or Gap*, $\mathrm{GE} : \mathscr{H} \to \mathbb{R}^+ \cup \{0\}$ and is defined as

$$\mathrm{GE}(\mathbf{H}) := |\mathcal{R}_\mu(\mathbf{H}) - \mathcal{R}_{\mu_N}(\mathbf{H})|. \tag{4}$$

Our aim is to derive non-asymptotic probabilistic upper bounds for the random variable $\mathrm{GE}(\hat{\mathbf{H}}_N)$ that depend explicitly on $n_x$, $\mathscr{H}$, and $N$.

**Existing approaches to generalization theory.** Before delving into the details of our main results, we first discuss several commonly used approaches to generalization theory and highlight their limitations in the context of GNNs. Broadly speaking, generalization bounds can be categorized into the following paradigms:

- *Classical uniform concentration*: This framework aims to compute the generalization error in the worst-case scenario, in other words, over the entire parameter space. The treatment of the data distribution varies as follows:

  - *VC theory* (Vapnik & Chervonenkis, 1971): One of the earliest and most classical approaches, VC theory, provides uniform guarantees over the hypothesis class and all distributions. However, computing VC dimensions for general neural networks is NP-HARD (Kranakis et al., 1995), and the resulting bounds are often vacuous.
  - *Distribution specific* (Shalev-Shwartz et al., 2009): In contrast to VC theory, these approaches tighten the bounds by restricting attention to specific classes of data distributions that are close to practical settings (e.g., sub-Gaussian). While this leads to improved generalization estimates, the analysis still relies on uniform concentration over the entire parameter space.

- *Geometric Analysis* (Hochreiter & Schmidhuber, 1997): These connect the geometric properties of empirical risk landscape near found solutions with generalization capability. However, such connections are not always necessary, and counterexamples exist (Dinh et al., 2017; Mulayoff & Michaeli, 2020).

- *PAC-Bayes Bounds* (McAllester, 1998): These adopt a Bayesian perspective, requiring a prior belief over the parameter space, and a posterior observation once training data is seen. The generalization error is then bounded via the KL-divergence between the prior and the posterior, which is typically intractable to compute for general networks.

- *Rademacher and Gaussian complexity* (Bartlett & Mendelson, 2002): These capture the ability of models to fit noise and act as a proxy for bounding generalization error. However, they are difficult to compute for deep architectures without strong assumptions.

- *Algorithmic stability* (Bousquet & Elisseeff, 2002): These bounds measure the stability of learning algorithms under perturbations to the training data. While potentially tight, the stability of commonly used algorithms such as stochastic gradient descent (SGD), ADAM (Kingma & Ba, 2015) does not always hold true for generic networks (Zhang et al., 2022).

- *Information theory* (Hellström et al., 2025): These bounds infer about the generalization error by quantifying the mutual information between the learned parameters and the training data. While these bounds can be tight, they are often infeasible to compute in deep learning settings due to similar issues as that of PAC-Bayes.

**Limitations of prior approaches.** Given the trade-offs of the aforementioned approaches, we focus on a classical generalization error bound from Shalev-Shwartz et al. (2009), which is based on uniform concentration over the parameter space under a fixed bounded data distribution $\mu$. We present this result as a corollary (see §A.6 for proof) and then point out the limitations of this setting in the context of single-layered, and single-channeled GCN.

**Corollary 1.** *Let $L = 1$, $\mathscr{H} \subset \mathbb{B}_{n_f}(B)$, and support of $\mu$ be bounded by $G$. Then w.p. of at least $1 - \delta$ it holds that*

$$\sup_{\mathbf{H} \in \mathscr{H}} \mathrm{GE}(\mathbf{H}) \leq \mathcal{O}\left(GB\sqrt{\frac{n_f \ln(N) \ln(n_f/\delta)}{N}}\right). \tag{5}$$

**Discussion.** For a single layer GCN, the bound in Equation (5) scales with the total number of trainable filter coefficients, $n_f$. In the limit when $n_f \to \infty$, the upper bound grows unboundedly, which is undesirable. This is particularly problematic in over-parameterized regimes. However, it is worth noticing the set of functions that can be represented by this GCN is significantly smaller due to the convolutional structure of the layer (see Equation 1). Specifically, since the graph signals are finite-length vectors, the Fourier transform is band-limited and can be represented by $\mathbb{C}^{n_x}$ (see Oppenheim (1999)).

At the crux of proof techniques for uniform concentration bounds lies the computation of hypothesis class complexity, measured by the covering number (to be defined precisely later), which represents the minimal number of hypotheses required to accurately represent the entire hypothesis class under a specified error tolerance. Since network parameters are directly used to construct the hypothesis class, the covering number scales exponentially with the number of parameters, $n_f$, leading to the bound in Equation (5). However, Fourier theory provides an equivalence between filter coefficients and their spectral representation. Here, we exploit this structure to represent the hypothesis class in terms of the spectral representation of the filters. This makes the covering numbers scale exponentially with the number of nodes, $n_x$, rather than parameters, leading to tighter bounds in over-parameterized regimes.

## 3 MAIN RESULT

In this section, we present upper bounds for generalization error for GCNs. In §3.1, we state and discuss the standing assumptions that are required for our main results to hold. Later, in §3.2, our main theorem for GCNs is presented. Finally, in §3.3 we apply our main theorem in various regimes of interest.

### 3.1 Standing Assumptions

Here, we introduce standing assumptions, and necessary tools. First, we assume that the unknown data distribution $\mu$ belongs to sub-Gaussian family.

**Assumption 1.** *The input signal* $\mathbf{x}$ *is a non-degenerate sub-Gaussian vector with proxy variance* $\sigma^2$*; i.e., for any* $\mathbf{a} \in \mathbb{S}^{n_x-1}$ *and for all* $t \geq 0$*, we have*

$$\mathbb{E}\left[\exp\left(t\langle \mathbf{x} - \mathbb{E}[\mathbf{x}], \mathbf{a}\rangle\right)\right] \leq \exp\left(t\sigma^2/2\right), \ \ and \ \ \mathbb{E}\left[\|\mathbf{x}\|_2^2\right] > 0.$$

*The target signal takes the form* $\mathbf{y} = g(\mathbf{x}) + \epsilon$*, where* $g : \mathbb{R}^{n_x} \to \mathbb{R}^{n_y}$ *is* $L_g$*-Lipschitz continuous function, and* $\epsilon$ *is an independent sub-Gaussian vector with proxy variance* $\sigma_\epsilon^2$*. As a result, the joint distribution* $\mu$ *is part of sub-Gaussian family.*

Sub-Gaussian models encompass a wide range of practical scenarios, including bounded distributions, Gaussian distributions (and their mixtures), beta and Dirichlet families, among others. This assumption is mild and widely adopted in the statistical learning theory literature (Pensia et al., 2018; Cao et al., 2021; Tadipatri et al., 2025). In GCN literature, boundedness of data is often assumed (Liao et al., 2020), making our assumption general.

Next, we assume certain regularity condition on loss and activation function.

**Assumption 2.** *The loss* $\ell$ *is convex and* $\zeta$ *smooth w.r.t the second argument. The activation function in each layer is* 1*-Lipschitz continuous.*

The convexity and smoothness condition on loss is a very common assumption in the literature. Lipschitz continuity on activation plays a pivotal role in our theoretical analysis similar to Shalev-Shwartz et al. (2009), as it enables sharp concentration guarantees when the inputs are drawn from sub-Gaussian distributions. Importantly, this assumption is not strictly necessary for concentration: weaker conditions can also yield such bounds, albeit with slower rates (Adamczak & Wolff, 2015). Commonly used activations such as ReLU, sigmoid, tanh, and softmax follow Lipschitz continuity (Gao & Pavel, 2017).

**Spectral representation.** Foundational blocks of GCNs are the graph filters and they admit a spectral representation. Since the convolutional operators are symmetric, they can be diagonalized as $S = V\Lambda V^{\mathsf{H}}$. The matrix $\Lambda \in \mathbb{C}^{n_x}$ whose diagonal entries are referred to as graph spectrum. The Graph Fourier Transform (GFT) for any signal $\mathbf{z} \in \mathbb{R}^{n_x}$ is defined as $\mathcal{F}\mathbf{z} := V^{\mathsf{H}}\mathbf{z} = \hat{\mathbf{z}} \in \mathbb{C}^{n_x}$, and similarly the inverse GFT is defined as $\mathcal{F}^{\dagger}\hat{\mathbf{z}} = \mathbf{z}$. By applying GFT on both the sides of Equation (1) we obtain

$$\mathcal{F}\phi(\mathbf{h}; \mathbf{z}) = \sum_{k=1}^{n_f} h_k \Lambda^k V^{\mathsf{H}} \mathbf{z} = \left(\sum_{k=1}^{n_f} h_k \Lambda^k\right) \mathcal{F}\mathbf{z}, \tag{6}$$

from which we have the spectral representation of graph filter / spectra as $\tilde{h}(\lambda) = \sum_{k=1}^{n_f} h_k \lambda^k$. For future use we denote spectra as

$$\mathcal{F}\mathcal{H} := \left\{ \sum_{k=1}^{n_f} h_k \mathsf{diag}^{\dagger}\left(\Lambda^k\right) : \forall \{h_k\} \in \mathcal{H} \right\} \subseteq \mathbb{C}^{n_x}, \tag{7}$$

and likewise the set $\mathcal{F}\mathscr{H}$ denotes the spectra of all layers and channels.

Finally, we introduce the notion of covering numbers that will be used in our main results.

**Definition 1** ($\varepsilon$-Covering Number (Vershynin, 2018))**.** *Let* $\mathcal{A}$ *be a set equipped with a semi-metric* $d$*. An* $\varepsilon$*-net of set* $\mathcal{A}$*, denoted by* $\mathcal{C}(\mathcal{A}, d, \varepsilon)$*, is any set of points* $\{h'_k\} \subseteq \mathcal{A}$ *such that every point* $h \in \mathcal{A}$ *lies within distance* $\varepsilon$ *of some* $h'_k$*; i.e.,*

$$\mathcal{C}(\mathcal{A}, d, \varepsilon) := \{h'_k\} : \forall h \in \mathcal{A}, \ \exists h'_k \in \mathcal{A} \ such \ that \ d(h, h'_k) \leq \varepsilon.$$

*The* $\varepsilon$*-covering number of* $\mathcal{A}$*, denoted by* $\mathcal{N}(\mathcal{A}, d, \varepsilon)$ *is the minimal cardinality of* $\mathcal{C}(\mathcal{A}, d, \varepsilon)$*. The natural logarithm of covering number is called the* metric entropy *of the set* $\mathcal{A}$*.*

### 3.2 A Sharp Generalization Bound for GCNs

With the above assumptions and definitions in place, We now present the main generalization error bound. To avoid notational clutter, we omit few deterministic constants, and proofs which can be found in the §A.4, and §A.5 respectively.

**Theorem 1.** *Under the Assumptions 1, and 2. Suppose $\{(\mathbf{x}_i, \mathbf{y}_i)\}_{i \in [N]}$ are i.i.d. samples drawn from the distribution $\mu$ and $\mathcal{H}$ be a compact set. Define the quantities*

$$L_{\mathcal{X}} := \left( \prod_{l \in [L]} C_l \sup_{\mathbf{h} \in \mathcal{H}} \|\mathcal{F}_l \mathbf{h}\|_2 \right) \quad , L_{\mathcal{H}} := \left[ \mathbb{E}\left[ \|\mathbf{x}\|_2 \right] + \sup_{l \in [L], \mathbf{h} \in \mathcal{H}} \|\mathcal{F}_l \mathbf{h}\|_2 \right]$$

$$K := n_y \zeta \left[ (L_{\mathcal{X}}^2 + L_g^2)\sigma^2 + \sigma_e^2 \right], \quad K'' := \max\left\{ 2L_{\mathcal{H}} \left[ L_{\mathcal{H}} \Delta_{\|\cdot\|_2}(\mathcal{H}) + \Phi(0;0) \right], K+1 \right\}.$$

*Fix a $\delta \in (0, 1]$. Then for any global minimizer $\hat{\mathbf{H}}_N$ of (ERM) w.p. of at least $1 - \delta$ we have*

$$\mathsf{GE}\left( \hat{\mathbf{H}}_N \right) \leq \inf_{\varepsilon \in (0, K]} \left( 2\varepsilon + K\sqrt{\frac{\ln(3/\delta) + \sum_{l \in [L]} C_l C_{l-1} \ln(\mathcal{N}(\mathcal{F}_l \mathcal{H}, \|\cdot\|_2, \varepsilon/K''))}{2N}} \right) \quad (8)$$

**Remarks.** The bound in Equation (16) parallels classical uniform concentration results for Lipschitz continuous functions under sub-Gaussian data distributions. We now have flexibility to substitute the covering number of class of spectra considered. The infimum can be easily upper bounded by certain choices of $\varepsilon$, like demonstrated in §3.3. Constants $L_{\mathcal{X}}$ (or $L_{\mathcal{H}}$) are the Lipschitz constants of the GCN w.r.t. the input data (or parameters). $K$ is the sub-Gaussian proxy variance (or an upper bound thereof) of the GCN's output and $K''$ is just a internal constant that arises in the proof.

**Proof outline.** Our proof technique relies on applying union bound for the tail probabilities of the empirical process GE, over the hypothesis class $\mathcal{H}$. This requires computing the covering number of $\mathcal{H}$ under the metric $d'(\mathbf{h}, \mathbf{h}') := \|\sum_{j \in [n_f]} (h_k - h'_k) S^k\|_2$. However, by Parseval theorem (Parseval, 1806), we can equivalently compute the metric $d'(\mathbf{h}, \mathbf{h}') = \|\mathcal{F}\mathbf{h} - \mathcal{F}\mathbf{h}'\|_2$. This equivalence is crucial, as it allows us to transfer from $\mathbb{R}^{n_f}$ to $\mathbb{C}^{n_x}$.

**Limitations.** (i) While our results require $\mu$ to belong to the sub-Gaussian class. Although we employ this for technical convenience, the GFT is still applicable. Extensions to heavy-tailed distributions are possible (Li et al., 2024), albeit with slower error rates.

(ii) It is not necessary to solve (ERM) exactly, our results extend to any first-order stationary points. This requires positive homogeneity of the network $\Phi(\mathbf{H}; \cdot)$ w.r.t its parameters, which enables connecting the nonconvex and convex programs (Tadipatri et al., 2025).

(iii) Finally, our bounds apply only when the convolutional operator $S$ is fixed. For GNNs that learn $S$ from data, such as Graph Attention Networks (Veličković et al., 2018; Franceschi et al., 2019), extending our results is non-trivial, since GFT is not uniform across draws from $\mu$. Here, prior art often resorts to classical frameworks (Vasileiou et al., 2025).

### 3.3 Escaping Over-parameterization in GCNs

In this section, we apply Theorem 1 to graphs in different regimes. In Corollary 2, we consider the case when graph is finite sized. In Corollary 3, and 4, we study the case of infinite sized graphs, or referred to as *graphons* (Ruiz et al., 2020).

Optimization algorithms such as SGD, ADAM to solve (ERM) often lead to parameters that are bounded (Reddy & Vidyasagar, 2023). For GCNs this implies that the spectrum is bounded, such phenomenon is also observed in Ruiz et al. (2020); Wang et al. (2025b). We now apply Theorem 1 under this boundedness condition.

**Corollary 2.** *Let various symbols be as in Theorem 1. If $\mathcal{F}\mathcal{H}$ is bounded by 1, then w.p. of at least $1 - \delta$ it holds that*

$$\mathrm{GE}\left( \hat{\mathbf{H}}_N \right) \leq K\sqrt{\frac{2Ln_x}{N} \ln\left( 1 + \max\left\{ \frac{4K''}{K}\sqrt{\frac{2N}{Ln_x}}, e - 1 \right\} \right)} + K\sqrt{\frac{\ln(3/\delta)}{2N}}. \quad (9)$$

**Remarks.** From Equation (9), we conclude that the sample complexity is $N \geq \tilde{\mathcal{O}}(Ln_x)$ and independent of the number of parameter $n_f$. Meanwhile, FCNNs with same number of

Table 1: Comparison of generalization error bounds for GCN.

| Model | Work | Technique | $\text{GE}(\hat{\mathbf{H}}_N) \leq \tilde{\mathcal{O}}(\cdot)$ |
|-------|------|-----------|-------------------------------------|
| FCNN | Bartlett et al. (2019) | VC-dimension | $\sqrt{((L-1)n_x^2 + n_y n_x)/N}$ |
| GCN | Scarselli et al. (2018) | VC-dimension | $\sqrt{L^2 n_f^4 n_x^2/N}$ |
| | Liao et al. (2020) | PAC-Bayes ($n_f = 2$) | $\sqrt{L^2 n_x/N}$ |
| | Garg et al. (2020) | Rademacher complexity | $\sqrt{Ln_x^3/N}$ |
| | **Ours** | Covering number | $\boldsymbol{\sqrt{Ln_x/N}}$ |

layers and hidden dimensions would require $N \geq \tilde{\mathcal{O}}((L-1)n_x^2 + n_y n_x)$ (Bartlett et al., 2019). Ours bound provide theoretical evidence for the empirical success of GCNs over FCNNs.

**Comparison with state-of-the-art bounds.** Our generalization bounds are not only independent of the number of parameters, but also outperform existing state-of-the-art bounds for GCNs (see Table 1). Scarselli et al. (2018) employ VC-theory framework, their analysis yields a sample complexity of $N \geq \tilde{\mathcal{O}}\left(L^2 n_f^4 n_x^2\right)$, which scales poorly with $L$, $n_f$ and $n_x$. Liao et al. (2020) obtained slightly tighter bound $N \geq \tilde{\mathcal{O}}\left(L^2 n_x\right)$ by using PAC-Bayes framework but they consider GCN with second order graph filters (Kipf & Welling, 2017). This bound scales linearly with the $L$, and also makes it less effective in simple cases when trying to predict low-pass graph signals, which require countably infinite number of filter coefficients (see Oppenheim (1999)). Garg et al. (2020) showed bounds that have no dependence on $n_f$, and require a sample complexity of $N \geq \tilde{\mathcal{O}}\left(Ln_x^3\right)$ by using Rademacher complexity. However, cubic dependence on $n_x$ makes it loose compared to our bound. Tang & Liu (2023) provides generalization bounds through algorithmic stability properties of SGD, but lacks explicit dependence on $n_f$ or $n_x$. Other works (Wang et al., 2025b;c), use the term "generalization bounds" for GCNs in a different context. Their settings consider the closeness of test performance to the best possible in expectation when the graph itself is generated by a fixed manifold, which is different from our setting. In comparison to other works, our bounds do not scale with $n_f$. To the best of our knowledge, these are the *first theoretical results* that analyze generalization properties in the spectral domain.

Now we extend our results to infinite node regime where the number of nodes $n_x \to \infty$, these graphs are called *graphons* (Ruiz et al., 2020). Graphons are very relevant to the study of network science (Vizuete et al., 2021), game theory (Parise & Ozdaglar, 2019), and controls (Gao & Caines, 2019b). In this regime, error bounds in Equation 9 quickly become vacuous. The poor scaling is often referred to as "curse of dimensionality", this term was first coined by Bellman (1954). However, the spectrum of graphons filters are known to exhibit certain regularity conditions (Ruiz et al., 2020). This curse of dimensionality can be avoided by imposing certain regularity conditions on the spectrum such as Lipschitz continuity or low-pass nature. We now apply Theorem 1 when the spectrum is also Lipschitz continuous.

**Corollary 3.** *Let various symbols be as in Theorem 1 with $C_l = 1$. If the each layer's hypothesis class $\mathcal{FH} \subseteq \{\{x_i\} : \forall i \in \mathbb{N}, f \in \mathcal{A}; x_i = f(\lambda_i)\}$, where $\mathcal{A} := \{f : \mathbb{C} \to \mathbb{C} : \forall \lambda, \lambda' \in \mathbb{C}, |f(\lambda) - f(\lambda')| \leq P|\lambda - \lambda'|, \|f\|_\infty \leq 1, f(0) = 0\}$. Then w.p. of at least $1 - \delta$ we have*

$$\text{GE}\left(\hat{\mathbf{H}}_N\right) \leq \mathcal{O}\left(\left(\frac{L^{1/6}P^{2/3}}{N^{1/6}}\right) + \sqrt{\frac{\ln(3/\delta)}{N}}\right). \tag{10}$$

**Remarks.** The error rate only scales as $N^{-1/6}$, which is slower than the central limit theorem rate of $N^{-1/2}$. However, the astounding aspect is that the dependence on $n_x$ and $n_f$ vanishes, effectively escaping the curse of dimensionality. Moreover, it has been well studied that Laplacian of graphons tend to have bounded eigenvalues (Gao & Caines, 2019a). Therefore, the spectrum implicitly satisfies the Lipschitz continuity condition.

Nt & Maehara (2019) observed that GCNs naturally learn low-pass filters for certain datasets. This suggests that spectral response is much more regular that just Lipschitz

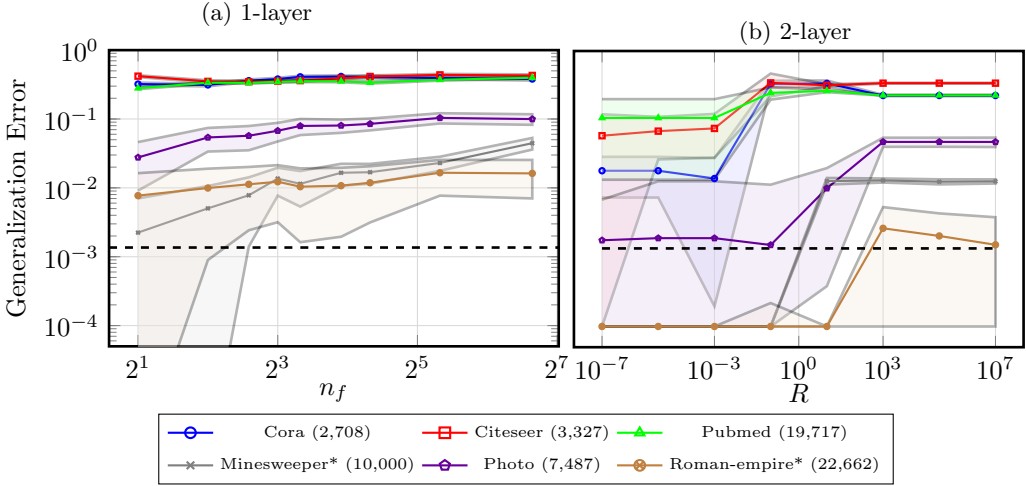

Figure 1: Effect of Lipshitz constant, and filter length on generalization error. Heterophilous datasets are marked with "*". Dashed line indicates the lowest testing standard error.

continuity. We formalize this regularity and call a spectrum is low-pass with bandwidth $\gamma$ and order $k$ if the magnitude of the spectra is of form

$$|\tilde{h}(\lambda)| \leq \begin{cases} A_{\mathsf{pass}} & \text{if } |\lambda| \leq \gamma, \\ A_{\mathsf{pass}}/|\lambda|^k & \text{otherwise.} \end{cases} \tag{11}$$

Next, we apply Theorem 1 when the spectrum is also low-pass.

**Corollary 4.** *Let various symbols be as in Theorem 1 with $L = 1$. If the hypothesis class $\mathcal{H}$ consists of low-pass spectrum with order $k > 1/2$. Then there are deterministic constants $\alpha_k, \beta_k > 0$ such that w.p. of at least $1 - \delta$ we have*

$$\mathrm{GE}(\hat{\mathbf{H}}_N) \leq \alpha_k \sqrt{\frac{\ln\left(1 + \max\left\{\beta_k (N/L)^{(2k-1)/4k}, e^{1/(2k-1)^2} - 1\right\}\right)}{(N/L)^{(2k-1)/2k}}} + K\sqrt{\frac{\ln(3/\delta)}{2N}}. \tag{12}$$

*Moreover, in the limit as $k \to \infty$ the above relation evaluates to*

$$\mathrm{GE}(\hat{\mathbf{H}}_N) \leq 8\sqrt{\frac{L\ln(1 + A_{\mathsf{pass}}\sqrt{8N/L})}{2N}} + K\sqrt{\frac{\ln(3/\delta)}{2N}}. \tag{13}$$

**Remarks.** For $k$th order low-pass spectrum, the generalization error scales as $N^{-(2k-1)/4k}$, as soon as $k \geq 0.75$ we obtain a faster rate than Corollary 3. In the limit when $k \to \infty$, we recover the best possible rate of $N^{-1/2}$. This indicates that enforcing low-pass filters is beneficial for generalization. However, it is not necessarily true that such constraints yield better expressive power or training performance.

## 4 NUMERICAL SIMULATIONS

In this section, we corroborate our insights by performing numerical simulations. First, to validate our theoretical upper bounds with empirical observations we design a synthetic experiment using Erdos-Réyi-Gilbert model. Finally, we consider several real-world datasets both homophilic (node features are similar when targets are similar) and heterophilic (node features are not similar when targets are similar) to perform node classification. The datasets include Cora, Citeseer, Pubmed (Yang et al., 2016), Photo, Computers (Shchur et al., 2018), Wikics (Mernyei & Cangea, 2020), Minesweeper, Tolokers, Roman-empire, Amazon-ratings, and Questions (Platonov et al., 2023).

In practice, it is computationally inefficient to implement graph filters of the form in Equation (1) directly, especially for large graphs. To address this, we parametrize graph filters

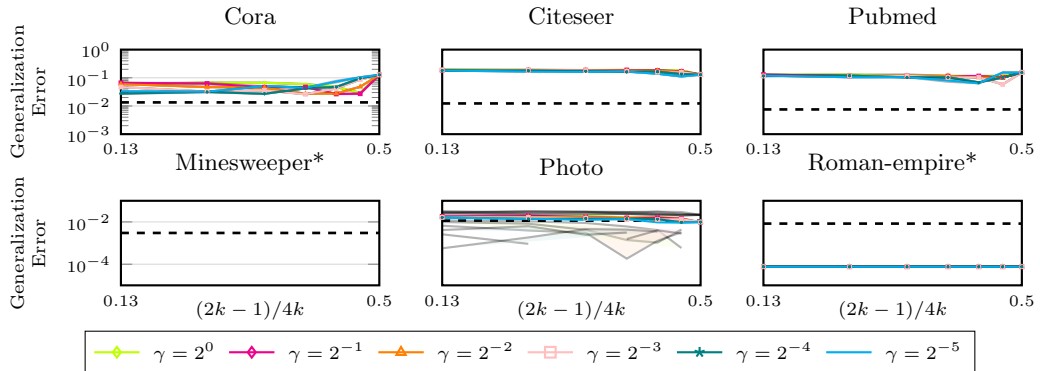

Figure 2: Effect of low-pass spectrum on the generalization error. $\gamma$ is band-width, and $k$ is exponent of spectrum decay. Heterophilous datasets are marked with "*". Dashed line indicates the lowest testing standard error.

using a Chebyshev polynomial basis, whose approximation properties are well-studied and known to be min–max optimal and unique in polynomial space (Geddes, 1978).

We adopt the off-the-shelf ChebNet architecture (Defferrard et al., 2016), which is based on Chebyshev polynomials, to implement the GCN layers. Our model uses a multi-layer ChebNet with ReLU activations and is trained using cross-entropy loss. We train for 200 epochs using the ADAM optimizer (Kingma & Ba, 2015) with a learning rate of 0.01. We provide experimental evidence to verify the theoretical bounds from Corollaries 2, 3, and 4. The results are averaged over 5 random seeds. Since the test set itself is random and finite-sampled, we ignore the generalization errors that are less than the standard deviation of the test error fluctuations, i.e., $\ln(n_c)\sqrt{1/2M}$, where $n_c$ is the number of classes, and $M$ is the number of test samples. In our simulations, datasets such as Computers, Wikics, Tolokers, Amazon-ratings, and Questions are not shown because the obtained generalization errors are of order $\approx 10^{-7}$, which is less than the standard deviation of the test error fluctuations, $\approx 10^{-3}$ (see §A.2).

**Validation of theoretical bounds.** In Figure 3 we plot the empirical and theoretical generalization errors on a synthetic dataset generated using the Erdos-Rényi-Gilbert model for various training sample and sizes. We observe that analyzed upper bounds are consistent with empirical observations. Moreover, the trends of $\mathcal{O}(1/\sqrt{N})$ are consistent with the theoretical bounds. However, there is seems to be a constant gap between the theoretical and empirical errors, which is expected since our bounds are catered to track the trends on the problem parameters but not necessarily tight in constants.

**Generalization performance is insensitive to the filter length.** Corollary 2 establishes that generalization error is independent of $n_f$. To verify this claim, we train a 1-layer ChebNet on different datasets while clipping the norm of the parameters to be less than $10^8$ for boundedness of the spectrum, with a varying number of filter taps, $n_f$. Figure 1a shows that the generalization error is relatively constant across large variations in $n_f$.

**Lower Lipschitz constant improves generalization.** To verify Corollary 3, we train a 2-layer ChebNet on different datasets via projected ADAM, at each iteration we project the parameters onto a Euclidean ball of desired radius $R$. This allows us to control the Lipschitz constant of the GCN, since the Lipschitz constant of the filter spectrum is directly proportional to the $\ell_2$ norm of the spectrum. In Figure 1b, we vary the $R$, and observe that lower $R$ (i.e., lower Lipschitz constant) yields better generalization performance consistently across datasets. This empirical evidence supports the upper bound in Equation 10.

**Low-pass spectrum yield better generalization.** To verify Corollary 4, we train a 1-layer ChebNet composed with a graph low-pass filter having varying bandwidth $\gamma$, and order $k$. Figure 2 shows that the generalization error on a log scale remains invariant to

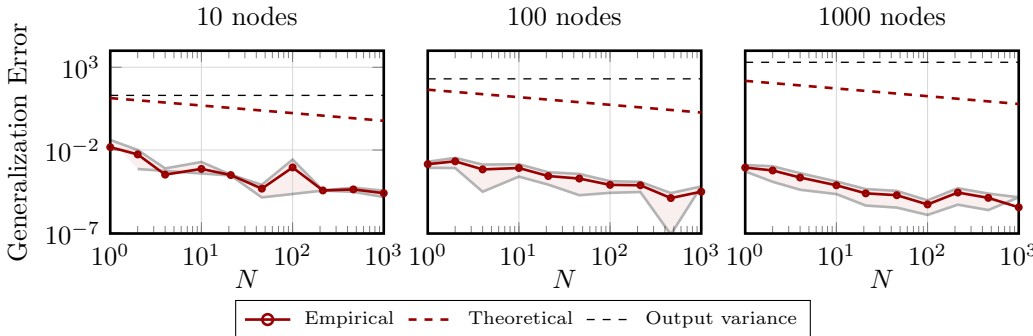

Figure 3: Verification of theoretical upper bounds on Erdos-Rényi-Gilbert model with connection probability 0.05 and node features sampled i.i.d. from $\mathcal{N}(0,1)$. The scalar targets are generated from a random two-layer ChebNet with unit-norm weights and ReLU activation. Generalization error is computed over $N$ training and 1000 test samples with different graph size.

$(2k - 1)/4k$, which is consistent with the exponent in the upper bound of Equation 12. Errors for the Minesweeper dataset is not visible because it is very close to zero.

## 5 Conclusions

In this work, we derived sharp generalization bounds for multi-layer, multi-channel GCNs by leveraging classical tools from signal processing and modern techniques in statistical learning theory. Unlike prior approaches that analyze GCNs purely through their parameter space, we adopt a spectral viewpoint of GCN convolutional layers, which admit lower intrinsic dimensionality. By exploiting this spectral structure, we derive generalization bounds that are independent of the total number of trainable parameters, and instead scale nearly linearly with the input dimension or number of nodes. In the finite-node setting, the sample complexity scales nearly linearly with the number of nodes. In the infinite-node setting, under certain mild regularity conditions on the filter spectrum, we show that GCNs provably escape the curse of dimensionality. Our theoretical findings are corroborated by extensive numerical simulations on real-world datasets.

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

# A   APPENDIX

In this supplementary material, we discuss the remainder of the proofs for mathematical statements made, extra technical discussion and related work. The below is the table of contents for the appendix.

# Contents

## A.1   EXTENSIONS TO CNNS

**Example 1.** *Let $X \in \mathbb{R}^{m \times n}$ be an image. We interpret the graph signal as $\mathbf{x} = \mathsf{vec}(X) \in \mathbb{R}^{m \cdot n}$, where $\mathsf{vec}(\cdot)$ is the vectorization of $X$ obtained by column-wise stacking.*

*Consider the pixel location $(a, b) \in [m] \times [n]$ the corresponding location in the vectorized image is $a + (b - 1)m$. Suppose that each pixel location $(a, b)$ is connected to its 8-neighbors (if exists) namely $(a - 1, b - 1)$, $(a - 1, b)$, $(a - 1, b + 1)$, $(a, b - 1)$, $(a, b + 1)$, $(a + 1, b - 1)$, $(a + 1, b)$, $(a + 1, b + 1)$. The corresponding locations of these pixels in the vectorized image are*

$$
\begin{aligned}
(a, b) &\to i := a + (b - 1)m \\
(a - 1, b - 1) &\to (a - 1) + (b - 2)m = i - (m + 1) \\
(a - 1, b) &\to (a - 1) + (b - 1)m = i - 1 \\
(a - 1, b + 1) &\to (a - 1) + (b)m = i + (m - 1) \\
(a, b - 1) &\to a + (b - 2)m = i - m \\
(a, b + 1) &\to a + (b)m = i + m \\
(a + 1, b - 1) &\to (a) + (b - 2)m = i - (m - 1) \\
(a + 1, b) &\to (a) + (b - 1)m = i + 1 \\
(a + 1, b + 1) &\to (a) + (b)m = i + (m + 1).
\end{aligned}
$$

*Effectively the adjacency matrix of the graph is a $m \cdot n \times m \cdot n$ matrix with 8-neighborhood structure. The adjacency matrix $A$ is given by*

$$A_{i,j} = \begin{cases} 1, & \text{if } |i - j| \in \{1, m-1, m, m+1\} \\ 0 & \text{otherwise.} \end{cases} \tag{14}$$

*Pictorial representation of such transformation for $3 \times 3$ image is show in Figure 4.*

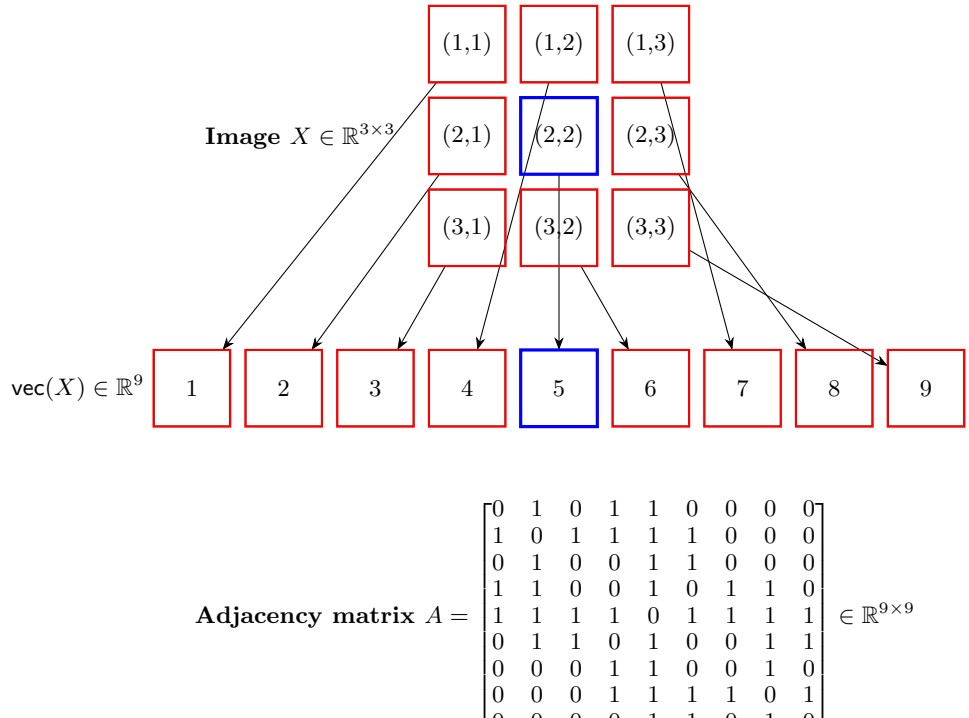

Figure 4: $3 \times 3$ image $X$ with symbolic pixel locations $(a, b)$, its vectorized form $\mathsf{vec}(X)$, and 8-neighborhood (if exists) adjacency matrix $A$.

From the above example, we can see that the any image can be represented as a graph signal $\mathbf{x} \in \mathbb{R}^{m \cdot n}$, where $m$ and $n$ are the number of rows and columns of the image respectively. Therefore, our results also apply to CNNs.

## A.2 Extra Numerical Simulations

We describe the dataset details in Table 2. We demonstrate both the test error and the generalization error for 2-layer and 10-layer ChebNet in Figure 5. To run our experiments, we use the PyTorch Geometric library for implementing GCNs.

In §A.2.1, we discuss the error in estimating the expected risk using empirical test samples.

### A.2.1 Empirical test samples

In practice, we do not have access to the test samples. Therefore, we cannot compute the expected risk to verify the closeness of the empirical risk and the expected risk. Instead, we use a held-out data to estimate the mean. This introduces a small error in the estimation of the expected risk. In the experiments, we consider node classifications tasks with cross-entropy loss, therefore the loss random variables always lies in $[0, \ln(n_c)]$, where $n_c$ is the number of classes. Therefore, we can use Hoeffding's inequality (Hoeffding, 1963) to bound the error in the estimation of the expected risk.

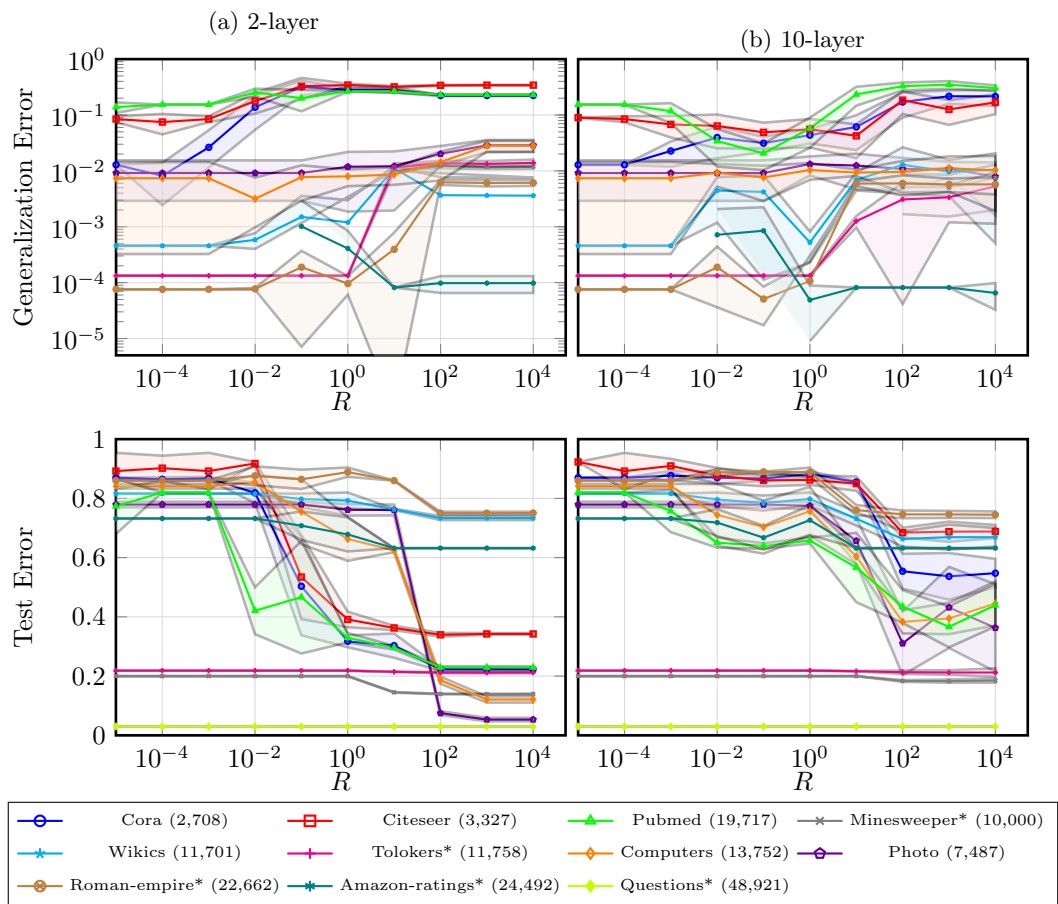

Figure 5: Dataset comparison showing performance across different graph datasets and configurations.

Table 2: Graph datasets used in numerical experiments with their characteristics. $\hat{\beta}$ is the regression coefficient of the linear fit between $\log(\mathbb{P}(\|\mathbf{x} - \mathbb{E}[\mathbf{x}]\|_2 > t))$ and $t^2$. Negative $\hat{\beta}$ indicates sub-Gaussian tails.

| Dataset | Nodes | Edges | Features | Classes | Homophily | Description | $\left(\max_i \mathbf{x}_i, \min_i \mathbf{x}_i, \hat{\beta}\right)$ | $1 - \mathbb{E}[\|\mathbf{x}\|_0]/n_x$ | $1 - |\mathcal{E}|/|\mathcal{V}|^2$ |
|---|---|---|---|---|---|---|---|---|---|
| Citeseer | 3,327 | 9,104 | 3,703 | 6 | Yes | Citation network | (1.0, 0.0, 0.0) | 0.991 | 0.999 |
| Cora | 2,708 | 10,556 | 1,433 | 7 | Yes | Citation network | (1.0, 0.0, 0.0) | 0.987 | 0.999 |
| Pubmed | 19,717 | 88,648 | 500 | 3 | Yes | Citation network | (1.26, 0.0, -7.29) | 0.900 | 0.9997 |
| Amazon-computers | 13,752 | 491,722 | 767 | 10 | Yes | Product co-purchase network | (1.0, 0.0, 0.0) | 0.652 | 0.994 |
| Amazon-photo | 7,487 | 119,043 | 765 | 8 | Yes | Product co-purchase network | (1.0, 0.0, 0.0) | 0.653 | 0.996 |
| Amazon-ratings | 24,492 | 93,050 | 300 | 5 | No | Product rating prediction | (1.87, -1.25, -3.48) | 0.000 | 0.9997 |
| Minesweeper | 10,000 | 39,402 | 7 | 2 | No | Mine detection in grid | (1.0, 0.0, 0.0) | 0.857 | 0.999 |
| Questions | 48,921 | 153,540 | 301 | 2 | No | Question classification | (3.36, -1.66, -1.08) | 0.152 | 0.9999 |
| Roman-empire | 22,662 | 32,927 | 300 | 18 | No | Historical network | (2.08, -1.76, -2.29) | 0.003 | 0.9999 |
| Tolokers | 11,758 | 519,000 | 10 | 2 | No | Worker classification | (1.0, 0.0, -4.39) | 0.480 | 0.992 |
| Wikics | 11,701 | 216,123 | 300 | 10 | Unknown | Wikipedia CS pages | (2.49, -2.05, -1.74) | 0.000 | 0.997 |

**Proposition 1.** *Let $\ell$ be a cross-entropy loss with $n_c$ classes and $\mu'_M$ be the empirical distribution of the test set with $M$ samples drawn i.i.d from $\mu$. Suppose that $\hat{\mathbf{H}}_N$ is independent from $\mu'_M$. Fix a $\delta \in (0, 1]$. Then w.p. of at least $1 - \delta$ we have*

$$\left| \mathcal{R}_{\mu'_M}(\hat{\mathbf{H}}_N) - \mathcal{R}_{\mu}(\hat{\mathbf{H}}_N) \right| \leq \ln(n_c) \sqrt{\frac{\ln(2/\delta)}{2M}}. \tag{15}$$

## A.3 EXISTING APPROACH

Table 1 summarizes existing generalization bounds for GCNs.

**Uniform concentration of measure.** The analysis in Shalev-Shwartz et al. (2009) is typically based on uniform concentration inequalities for Lipschitz continuous functions. Since the learnt parameters $\hat{\mathbf{H}}_N$ are correlated with the empirical distribution $\mu_N$ classical concentration arguments do not directly apply, that is, closeness of empirical risk to population risk. Instead, one must control the deviation uniformly over the parameter space $\mathscr{H}$; i.e., $\mathrm{GE}(\hat{\mathbf{H}}_N) \leq \sup_{\mathbf{H} \in \mathscr{H}} \mathrm{GE}(\mathbf{H})$ *a.s.* By the monotonicity of probability measures under inclusion, for any $\varepsilon \in \mathbb{R}$ we have

$$\mathbb{P}\left(\mathrm{GE}(\hat{\mathbf{H}}_N) \geq \varepsilon\right) \leq \mathbb{P}\left(\sup_{\mathbf{H} \in \mathscr{H}} \mathrm{GE}(\mathbf{H}) \geq \varepsilon\right)$$

To upper bound the right-hand side, one needs a uniform control over the generalization error across the entire parameter space. A key idea is to relate this to the "size" of the parameter space, which is captured by the notion of *covering numbers* (Dudley, 1974; Vershynin, 2018).

For instance, suppose the generalization error $\mathrm{GE}(\mathbf{H})$ is $K$-Lipschitz continuous with respect to $\mathbf{H}$ under the semi-metric $d$ a.s., that is, for all $\mathbf{H}_1, \mathbf{H}_2 \in \mathscr{H}$: $|\mathrm{GE}(\mathbf{H}_1) - \mathrm{GE}(\mathbf{H}_2)| \leq K \cdot d(\mathbf{H}_1, \mathbf{H}_2)$. Then via few simple algebraic manipulations we can show that

$$\mathbb{P}\left(\sup_{\mathbf{H} \in \mathscr{H}} \mathrm{GE}(\mathbf{H}) \geq \varepsilon\right) \leq \mathbb{P}\left(\sup_{\mathbf{H} \in \mathcal{C}(\mathscr{H}, d, \varepsilon)} \mathrm{GE}(\mathbf{H}) \geq (1+K)\varepsilon\right).$$

Since the right-hand side is a union over an $\varepsilon$-net, we can further bound it using the covering number and for some fixed $\mathbf{H}' \in \mathscr{H}$ independent from $\mu_N$, that is,

$$\mathbb{P}\left(\mathrm{GE}(\hat{\mathbf{H}}_N) \geq \varepsilon\right) \leq \mathcal{N}(\mathscr{H}, d, \varepsilon) \cdot \mathbb{P}\left(\text{for a fixed } \mathbf{H}' \in \mathscr{H} : \mathrm{GE}(\mathbf{H}') \geq (1+K)\varepsilon\right).$$

The probability term on the right can be upper bounded using standard concentration inequalities (e.g., sub-Gaussian). The tightness of the bound depends on the covering number $\mathcal{N}(\mathscr{H}, d, \varepsilon)$—smaller values yield tighter generalization bounds.

Suppose $\mathscr{H} \subset \mathbb{R}^{n_f}$ (i.e., $L = 1$) then the covering number for $p \geq 1$ we have $\mathcal{N}(\mathscr{H}, \|\cdot\|_p, \varepsilon) \leq \tilde{\mathcal{O}}(1/\varepsilon^{n_f})$, which roughly leads to the bound $\mathrm{GE}\left(\hat{\mathbf{H}}_N\right) \propto \sqrt{n_f/N}$. However, in the over-parameterized regime where $n_f \geq \mathcal{O}(n_x)$, this results in vacuous generalization bounds, highlighting the need for re-thinking the analysis.

### A.4 Constants

$\alpha_k = 4\left(\frac{K}{2(2k-1)}\sqrt{\frac{w_1}{2}}\right)^{(2k-1)/(2k)}$, and $\beta_k = w_2\left(\frac{2(2k-1)}{K}\sqrt{\frac{2}{w_1}}\right)^{1-1/2k}$, where $w_1 = 2\left(\frac{16A_{\mathsf{pass}}^2}{2k-1}\right)^{1/(2k-1)}$, and $w_2 = 4A_{\mathsf{pass}}K''\sqrt{\zeta(2k)}$, here $\zeta(\cdot)$ is the Riemann zeta function.

### A.5 Main proofs

In this section, we discuss the proof of Theorem 1, Before which we restate the general theorem that is applicable to multi-channel GNNs.

**Theorem 2.** *Under the Assumptions 1, and 2. Suppose $\{(\mathbf{x}_i, \mathbf{y}_i)\}_{i \in [N]}$ are i.i.d. samples drawn from the distribution $\mu$ and $\mathscr{H}$ be a compact set. Define the quantities*

$$L_{\mathcal{X}} := \left(\prod_{l \in [L]} C_l \sup_{\mathbf{h} \in \mathcal{H}} \|\mathcal{F}_l \mathbf{h}\|_2\right) \quad, L_{\mathscr{H}} := \left[\mathbb{E}\left[\|\mathbf{x}\|_2\right] + \sup_{l \in [L], \mathbf{h} \in \mathcal{H}} \|\mathcal{F}_l \mathbf{h}\|_2\right]$$

$$K := n_y \zeta \left[(L_{\mathcal{X}}^2 + L_g^2)\sigma^2 + \sigma_e^2\right], \quad K'' := \max\left\{2L_{\mathscr{H}}\left[L_{\mathscr{H}}\Delta_{\|\cdot\|_2}(\mathcal{H}) + \Phi(0; o)\right], K+1\right\}.$$

*Fix a $\delta \in (0, 1]$. Then for any global minimizer $\hat{\mathbf{H}}_N$ of (ERM) w.p. of at least $1 - \delta$ we have*

$$\mathrm{GE}\left(\hat{\mathbf{H}}_N\right) \leq \inf_{\varepsilon \in (0, K]}\left(2\varepsilon + K\sqrt{\frac{\ln(3/\delta) + \sum_{l \in [L]} C_l C_{l-1} \ln(\mathcal{N}(\mathcal{F}_l \mathcal{H}, \|\cdot\|_2, \varepsilon/K''))}{2N}}\right) \quad (16)$$

*Proof.* We use Lemma 6 to estimate the Lipschitz constants of the GCN function $\Phi$, denote the input Lipschitz constant by $L_\mathcal{X}$, and the parameter Lipschitz constant by $L_\mathscr{H}$. Now by invoking Lemma 2 with $K, K'$ we obtain

$$\mathbb{P}\left(\text{GE}(\hat{\mathbf{H}}_N) > \inf_{\varepsilon \in (0,K]} \left(2\varepsilon + K\sqrt{\frac{\ln(\mathcal{N}(\mathcal{F}\mathscr{H}, \sup \|\cdot\|_2, \varepsilon/\max\{K_1, K+1\})) + \ln(3/\delta)}{2N}}\right)\right)$$
$$\leq \delta.$$

By upper bounding $\ln(\mathcal{N}(\mathcal{F}\mathscr{H}, \sup \|\cdot\|_2, \varepsilon/\max\{K_1, K+1\}))$ by $\sum_{l\in[L]} C_l C_{l-1} \ln(\mathcal{N}(\mathcal{F}_l\mathcal{H}, \|\cdot\|_2, \varepsilon/\max\{K_1, K+1\}))$ and re-scaling the $\varepsilon$ we obtain the desired result. $\square$

First we re-state the uniform concentration of convex functions. In other words, the closeness of empirical loss and population loss for a given function map.

**Lemma 1** (Concentration of Convex loss (Tadipatri et al., 2025))**.** *Suppose the distribution $\mu$ satisfies the Assumption 1. Consider the estimators from the set of functions $f_\theta : \mathbb{R}^{n_x} \to \mathbb{R}^{n_y}$ as parameterized by $\theta \in \Theta$ that are $L_\Theta$-Lipschitz continuous with respective to inputs. Consider a loss function $\ell : \mathbb{R}^{n_y} \times \mathbb{R}^{n_y} \to \mathbb{R}$ that is convex and $\zeta$-smooth.*

*Let $\mathcal{C} \subseteq \mathbb{R}^{n_x}$ be some convex set independent of empirical samples $\{\mathbf{x}_i\}_{i\in[N]}$ that are drawn i.i.d from $\mu$ such that $\mathbb{P}(\forall i \in [N], X_i \in \mathcal{C}) \geq 1 - \delta_\mathcal{C}$. Define the quantities*

$$L^{(\mathcal{C})} := \sup_{\theta, \theta' \in \Theta, \mathbf{z} \in \mathcal{C}} \frac{\|f_\theta(\mathbf{z}) - f_{\theta'}(\mathbf{z})\|}{d(\theta, \theta')}, \quad B^{(\mathcal{C})} := \sup_{\theta \in \Theta, \mathbf{z} \in \mathcal{C}} \|f_\theta(\mathbf{z})\|, \tag{17}$$
$$K := n_Y \zeta \left[(L_\Theta^2 + L_g^2)\sigma_X^2 + \sigma_\epsilon^2\right], \text{ and}$$

$$B_{nrm}(\mathcal{C}) := \sup_{\theta, \theta' \in \Theta} \left|\mathbb{E}_{\mathbf{z}\sim\mu}\left[\|f_{\theta'} \circ \mathcal{P}_\mathcal{C}(\mathbf{z}) - f_\theta \circ \mathcal{P}_\mathcal{C}(\mathbf{z})\|_2^2\right] - \mathbb{E}_{\mathbf{z}\sim\mu}\left[\|f_{\theta'}(\mathbf{z}) - f_\theta(\mathbf{z})\|_2^2\right]\right| \tag{18}$$

*where $\mathcal{P}_\mathcal{C}(\cdot)$ denotes the Euclidean projection operator onto the set $\mathcal{C}$. If $L^{(\mathcal{C})}, B^{(\mathcal{C})}, K < \infty$, and $\hat{\theta}$. Then for any global minimizer of $\mathcal{R}_{\mu_N}$, $\epsilon \in (0, K]$ and some universal constant $c > 0$ we have*

$$\mathbb{P}\left(\left|\mathcal{R}_{\mu_N}(\hat{\theta}) - \mathcal{R}_\mu(\hat{\theta})\right| \geq \epsilon + B_{nrm}(\mathcal{C})\right) \leq 2\mathcal{N}(\Theta, d, \epsilon/(2L^{(\mathcal{C})}B^{(\mathcal{C})}))\exp\left(-cN\left(\epsilon/K\right)^2\right)$$
$$+ \delta_\mathcal{C}. \tag{19}$$

**Lemma 2.** *Under the settings of Lemma 1, suppose $f_0$ is constant with value $F_0$, and there exists positive constants $a$ and $b$ such that $L^{(\mathcal{C})} = a \sup_{\mathbf{z}\in\mathcal{C}} \|\mathbf{z}\| + b$. Define*

$$K_1 := 2\Delta_d(\Theta)\left(a\|\mathbb{E}[\mathbf{x}]\|_2 + b\right)^2 + 2F_0\left(a\|\mathbb{E}[\mathbf{x}]\|_2 + b\right).$$

*Then there exists a universal positive constant $c$ such that w.p. of at least $1 - \delta$ we have*

$$\left|\mathcal{R}_{\mu_N}(\hat{\theta}) - \mathcal{R}_\mu(\hat{\theta})\right| \leq \inf_{\varepsilon\in(0,K]}\left(2\varepsilon + K\sqrt{\frac{\ln\left(\mathcal{N}\left(\Theta, d, \varepsilon/\max\{K_1, 1+K\}\right)\right) + \ln\left(3/\delta\right)}{cN}}\right).$$

*Proof.* The proof involves invoking Lemma 1, and we break it into multiple steps. (I) We will choose a convex set $\mathcal{C}$ that contains the data points $\mathbf{x}$ with high probability. (II) Then we will bound the key constants such as Lipschitz constant $L^{(\mathcal{C})}$ and $B^{(\mathcal{C})}$. With these constants we will move on (III) to controlling the metric entropy, (IV) projection error $B_{nrm}(\mathcal{C})$, and (V) existence of data points in $\mathcal{C}$. Each of the earlier steps induces constraints on the choice of $\mathcal{C}$. In (VI) we will show that a good choice of $\mathcal{C}$ exists that satisfies all the constraints, obtaining the final bound.

**(I) Choosing $\mathcal{C}$:** Set $\mathcal{C} = \mathbb{E}[\mathbf{x}] + \mathbb{B}_2(z)$ for some $z > 0$.

**(II) Bounding key constants**: From our assumption we have

$$L^{(\mathcal{C})} = az + a\|\mathbb{E}[\mathbf{x}]\|_2 + b.$$

To bound the term $B^{(\mathcal{C})}$, on application of the triangle inequality we have

$$B^{(\mathcal{C})} \leq \sup_{\theta\in\Theta, \mathbf{z}\in\mathcal{C}} \|f_\theta(\mathbf{z}) - f_{\theta'}(\mathbf{z})\|_2 + \|f_0(\mathbf{z})\|_2 \leq L^{(\mathcal{C})}\Delta_d(\Theta) + \sup_{\mathbf{z}\in\mathcal{C}} \|f_0(\mathbf{z})\|_2.$$

**(III) Controlling the metric entropy**: Combing the above two inequalities we have

$$2L^{(\mathcal{C})}B^{(\mathcal{C})} \leq 2\Delta_d(\Theta)\left[L^{(\mathcal{C})}\right]^2 + 2F_0 L^{(\mathcal{C})}.$$

By plugging the choice of $\mathcal{C}$ we have

$$2L^{(\mathcal{C})}B^{(\mathcal{C})} \leq 2a^2\Delta_d(\Theta)z^2 + (4a\Delta_d(\Theta) + 2F_0)(a\|\mathbb{E}[\mathbf{x}]\|_2 + b)z$$
$$+ 2\Delta_d(\Theta)(a\|\mathbb{E}[\mathbf{x}]\|_2 + b)^2 + 2F_0(a\|\mathbb{E}[\mathbf{x}]\|_2 + b),$$

Define $A_1 := 2a^2\Delta_d(\Theta)$, and $B_1 := (4a\Delta_d(\Theta) + 2F_0)(a\|\mathbb{E}[\mathbf{x}]\|_2 + b)$. For some arbitrary $\gamma > 0$ and $\alpha \in (0,1)$ for the relation $\epsilon/2L^{(\mathcal{C})}B^{(\mathcal{C})} \geq \epsilon^{1-\alpha}/\gamma^\alpha$ to hold true, the following inequality must hold

$$(\epsilon\gamma)^\alpha \leq A_1 z^2 + B_1 z + K_1,$$

as $z > 0$, it is necessary and sufficient for $z$ to satisfy

$$0 < z \leq z_1(\epsilon, \gamma, \alpha) := \frac{B_1}{2A_1}\left[\sqrt{1 + \frac{(\epsilon\gamma)^\alpha - K_1}{B_1^2}} - 1\right]. \tag{20}$$

The admissible $\epsilon$ is when $B_1^2 > 4A_1(C_1 - (\epsilon\gamma)^\alpha) \equiv \epsilon > \frac{1}{\gamma}K_1^{1/\alpha}$.

$z_1(\epsilon, \gamma, \alpha)$ is a continuous function in its arguments, a increasing function of $\epsilon, \gamma$. In the case when $\epsilon\gamma \geq 1$, it is increasing in $\alpha$ and decreasing in $\alpha$ otherwise.

**(IV) Controlling the projection error**: From Corollary 5 we have that

$$B_{nrm}(\mathcal{C}) \leq C_1\sigma^2(a\sigma + a\|\mathbb{E}[\mathbf{x}]\|_2 + b)L_\Theta\Delta_d(\Theta)\exp\left(-C_2 z^2/\sigma^2\right).$$

Therefore, for $B_{nrm}(\mathcal{C}) \leq \epsilon$ to hold true, it is sufficient for

$$z \geq z_2(\epsilon, \gamma, \alpha) := \frac{\sigma^2}{C_2}\ln\left(\frac{C_1\sigma^2(a\sigma + a\|\mathbb{E}[\mathbf{x}]\|_2 + b)L_\Theta\Delta_d(\Theta)}{\epsilon}\right). \tag{21}$$

The function $z_2$ is continuous in its arguments, is strictly decreasing in $\epsilon$, and constant in $\gamma$ and $\alpha$.

**(V) Probability of existence of x in $\mathcal{C}$**: Since $\mathbf{x}$ is sub-Gaussian distribution for some universal constant $c > 0$ we have

$$\mathbb{P}(\forall i \in [N], X_i \notin \mathcal{C}) \leq 2\exp\left(-cNz^2/\sigma^2\right).$$

Therefore, it is sufficient to choose $z$ such that

$$\exp\left(-cNz^2/\sigma^2\right) \leq \exp\left(\ln\left(\mathcal{N}(\Theta, d, \epsilon/(2L^{(\mathcal{C})}B^{(\mathcal{C})}))\right) - cN(\epsilon/K)^2\right),$$

when $z \leq z_1$ we have that

$$\exp\left(-cNz^2/\sigma^2\right) \leq \exp\left(\ln\left(\mathcal{N}(\Theta, d, \epsilon/(2L^{(\mathcal{C})}B^{(\mathcal{C})}))\right) - cN(\epsilon/K)^2\right),$$
$$\leq \exp\left(\ln\left(\mathcal{N}(\Theta, d, \epsilon^{1-\alpha}/\gamma^\alpha)\right) - cN(\epsilon/K)^2\right).$$

Therefore, we have that

$$z \geq z_3(\epsilon, \gamma, \alpha, N) := c_3\sigma\sqrt{\max\left\{\frac{\epsilon^2}{K^2} - \frac{1}{N}\ln\left(\mathcal{N}(\Theta, d, \epsilon^{1-\alpha}/\gamma^\alpha)\right), 0\right\}}. \tag{22}$$

The function $z_3$ is continuous in its arguments, is increasing in $\epsilon$, decreasing in $\gamma$, increasing in $\alpha$ when $\epsilon\gamma < 1$, and decreasing in $\alpha$ when $\epsilon\gamma > 1$.

**(VI) Existence of good $z$:** We claim that there exists a good $z$ such that $z_1 \geq z \geq \max\{z_2, z_3\}$ with a proof presented next. With such a good choice of $z$ we have that

$$\mathbb{P}\left(\left|\mathcal{R}_{\mu_N}(\hat{\theta}) - \mathcal{R}_\mu(\hat{\theta})\right| \geq 2\epsilon\right) \leq 3\exp\left(\ln\left(\mathcal{N}(\Theta, d, \epsilon^{1-\alpha}/\gamma^\alpha)\right) - cN(\epsilon/K)^2\right). \tag{23}$$

**Proof of the (VI):** We will consider the equilibrium point of $z_2$ and $z_3$ for finite $N$, and in the limit as $N \to \infty$, that is, we consider the programs

$$\underline{\epsilon}^{(\gamma, \alpha, N)} := \inf_{\epsilon > 0} \epsilon \text{ s.t. } z_2(\epsilon, \gamma, \alpha) = z_3(\epsilon, \gamma, \alpha, N),$$

$$\epsilon_* := \inf_{\epsilon > 0} \epsilon \text{ s.t. } z_2(\epsilon, \gamma, \alpha) = \lim_{N \to \infty} z_3(\epsilon, \gamma, \alpha, N) = c_3 \sigma \epsilon / K.$$

Clearly $\epsilon_*$ is independent of $\gamma$ and $\alpha$, because in the limit the metric entropy term vanishes. From the definition of $z_3$ we have that $z_3(\epsilon, \gamma, \alpha, N) \leq c_3 \sigma \epsilon / K$, and $z_2$ does not depend on $N$. Therefore, it is true that $\epsilon_* \leq \underline{\epsilon}^{(\gamma, \alpha, N)}$ for any $N > 0$.

Now we find a feasible candidate for $\gamma$, consider the program

$$\gamma_* := \inf_{\gamma > 0} \gamma \text{ s.t. } z_1(\epsilon_*, \gamma, \alpha) \geq z_2(\epsilon_*, \gamma, \alpha) \geq z_3(\epsilon_*, \gamma, \alpha, N).$$

From earlier dicussion we know that in terms of $\gamma$, $z_1$ is increasing, $z_2$ is constant, and $z_3$ is decreasing. Moreover, $z_1$ can grow unboundedly with $\gamma$ and $z_2$ is constant. Therefore, $\gamma_*$ certainly exists and is finite. Recall from Equation (23) that for any arbitrary $\beta \geq 0$ and $z \in \left[z_2\left(\epsilon_*, N^\beta \gamma_*, \alpha\right), z_1\left(\epsilon_*, N^\beta \gamma_*, \alpha\right)\right]$, we have that

$$\mathbb{P}\left(\left|\mathcal{R}_{\mu_N}(\hat{\theta}) - \mathcal{R}_\mu(\hat{\theta})\right| \geq 2\epsilon\right) \leq \exp\left(\ln\left(3 \cdot \mathcal{N}(\Theta, d, \epsilon/(\epsilon\gamma_* N^\beta)^\alpha)\right) - cN\left(\epsilon/K\right)^2\right).$$

Now for $\epsilon \in \left[\frac{1}{\gamma_* N^\beta} K_1^{1/\alpha}, K\right]$ and by instantiating Lemma 1 we have that

$$\mathbb{P}\left(\left|\mathcal{R}_{\mu_N}(\hat{\theta}) - \mathcal{R}_\mu(\hat{\theta})\right| \geq 2\epsilon\right) \leq \exp\left(\ln\left(3 \cdot \mathcal{N}(\Theta, d, \epsilon/(K\gamma_* N^\beta)^\alpha)\right) - cN\left(\epsilon/K\right)^2\right), \quad (24)$$

the metric entropy was upper bounded due its non-increasing nature in $\epsilon$.

**Failure rate flipping**: Let $\delta$ be such that

$$\exp\left(\ln\left(3 \cdot \mathcal{N}(\Theta, d, \epsilon/(K\gamma_* N^\beta)^\alpha)\right) - c'N\left(\epsilon/K\right)^2\right) \leq \delta,$$

by re-arranging the terms we have that

$$0 \leq K\sqrt{\frac{\ln\left(3 \cdot \mathcal{N}(\Theta, d, \epsilon/(K\gamma_* N^\beta)^\alpha)\right) + \ln(1/\delta)}{cN}} \leq \epsilon.$$

Now we add $2\epsilon$ on both sides to obtain

$$2\epsilon \leq 2\epsilon + K\sqrt{\frac{\ln\left(3 \cdot \mathcal{N}(\Theta, d, \epsilon/(K\gamma_* N^\beta)^\alpha)\right) + \ln(1/\delta)}{cN}} \leq 3\epsilon. \quad (25)$$

The intermediate term in Equation (25) is a tight estimate in $\epsilon$, using this we obtain

$$\mathbb{P}\left(\left|\mathcal{R}_{\mu_N}(\hat{\theta}) - \mathcal{R}_\mu(\hat{\theta})\right| \geq 2\epsilon + K\sqrt{\frac{\ln\left(3 \cdot \mathcal{N}(\Theta, d, \epsilon/(K\gamma_* N^\beta)^\alpha)\right) + \ln(1/\delta)}{cN}}\right) \leq \delta.$$

Since the choices, $\alpha \in (0, 1)$ and $\beta \geq 0$ are arbitrary, we make good choices to obtain the tight rates as possible. We have the following cases:

**(a) $K + 1 > K_1$:** Define

$$\underline{\beta}(\alpha) := \frac{\ln(1 + K)/\alpha - \ln(K\gamma_*)}{\ln(N)},$$

Then we have that $\epsilon/(K\gamma_* N^{\underline{\beta}(\alpha)})^\alpha = \epsilon/(1 + K)$. Set $\underline{\alpha} := \max\left\{\frac{\ln(1+K)}{\ln(K\gamma_*)}, 1\right\}$, and $\beta = \underline{\beta}(\alpha)$, then for any $\alpha \leq \underline{\alpha}$, and $\epsilon \in \left[K\left(\frac{K_1}{1+K}\right)^{1/\alpha}, K\right]$ we have that

$$\mathbb{P}\left(\left|\mathcal{R}_{\mu_N}(\hat{\theta}) - \mathcal{R}_\mu(\hat{\theta})\right| \geq 2\epsilon + K\sqrt{\frac{\ln\left(\mathcal{N}(\Theta, d, \epsilon/(1 + K))\right) + \ln(3/\delta)}{cN}}\right) \leq \delta.$$

We choose a sequence $\{\alpha_k\}$ such that $\alpha_k \to 0$ and bounded away from 0, and $\underline{\alpha}$. Then we obtain the bound

$$\mathbb{P}\left(\left|\mathcal{R}_{\mu_N}(\hat{\theta}) - \mathcal{R}_\mu(\hat{\theta})\right| \geq \inf_{\epsilon \in (0,K]} \left(2\epsilon + K\sqrt{\frac{\ln\left(\mathcal{N}(\Theta, d, \epsilon/(1+K))\right) + \ln(3/\delta)}{cN}}\right)\right) \leq \delta.$$

**(b)** $K + 1 < K_1$**:** Define

$$\overline{\beta}(\alpha) := \frac{2\ln(K_1)/\alpha - \ln(K\gamma_*)}{\ln(N)},$$

Then we have that for all $\beta = \overline{\beta}(\alpha)$, $\epsilon/(K\gamma_* N^{\overline{\beta}(\alpha)})^\alpha = \epsilon/K_1$. Set $\overline{\alpha} := \max\left\{\frac{2\ln(K_1)}{\ln(K\gamma_*)}, 1\right\}$. Now we choose a sequence $\{\alpha_k\}$ such that $\alpha_k \to 0$ and bounded away from 0, and $\overline{\alpha}$. Then we obtain the bound

$$\mathbb{P}\left(\left|\mathcal{R}_{\mu_N}(\hat{\theta}) - \mathcal{R}_\mu(\hat{\theta})\right| \geq \inf_{\epsilon \in (0,K]} \left(2\epsilon + K\sqrt{\frac{\ln\left(\mathcal{N}(\Theta, d, \epsilon/K_1)\right) + \ln(3/\delta)}{cN}}\right)\right) \leq \delta.$$

This concludes the proof. $\qquad\square$

**Lemma 3.** *Suppose $\forall \theta \in \Theta, f_\theta$ is $L_\Theta$-Lipschitz continuous with respect to inputs, and $L^{(\mathbb{B}_2(\mathbf{x}))}$-Lipschitz continuous with respect to parameter for a fixed $\mathbf{x}$. Then the projection difference satisfies the inequality:*

$$B_{nrm}(\mathcal{C}) \leq 2L_\Theta \Delta_d(\Theta)\mathbb{E}\left[L^{(\mathcal{C})} + L^{(\mathbb{B}_2(\mathbf{x}))}\right] \mathbb{E}\left[\|\mathcal{P}_\mathcal{C}(\mathbf{x}) - \mathbf{x}\|_2\right].$$

*Proof.* Recall that the projection difference is defined as

$$
\begin{aligned}
B_{nrm}(\mathcal{C}) &= \sup_{\theta,\theta' \in \Theta} \left|\mathbb{E}\left[\|f_{\theta'} \circ \mathcal{P}_\mathcal{C} - f_\theta \circ \mathcal{P}_\mathcal{C}\|_2^2 - \|f_{\theta'} - f_\theta\|_2^2\right]\right| \\
&= \sup_{\theta,\theta' \in \Theta} \left|\mathbb{E}\left[\langle f_\theta \circ \mathcal{P}_\mathcal{C} - f_\theta - (f_{\theta'} \circ \mathcal{P}_\mathcal{C} - f_{\theta'}), f_\theta \circ \mathcal{P}_\mathcal{C} - f_{\theta'} \circ \mathcal{P}_\mathcal{C} + f_\theta - f_{\theta'}\rangle\right]\right| \\
&\leq \sup_{\theta,\theta' \in \Theta} \mathbb{E}\left[\|f_\theta \circ \mathcal{P}_\mathcal{C} - f_\theta - (f_{\theta'} \circ \mathcal{P}_\mathcal{C} - f_{\theta'})\|_2\right]\mathbb{E}\left[\|f_\theta \circ \mathcal{P}_\mathcal{C} - f_{\theta'} \circ \mathcal{P}_\mathcal{C} + f_\theta - f_{\theta'}\|_2\right] \\
&\leq \sup_{\theta,\theta' \in \Theta} \mathbb{E}\left[\|f_\theta \circ \mathcal{P}_\mathcal{C} - f_\theta\|_2 + \|f_{\theta'} \circ \mathcal{P}_\mathcal{C} - f_{\theta'}\|_2\right] \\
&\qquad\qquad \times \mathbb{E}\left[\|f_\theta \circ \mathcal{P}_\mathcal{C} - f_{\theta'} \circ \mathcal{P}_\mathcal{C}\|_2 + \|f_\theta - f_{\theta'}\|_2\right] \\
&\leq \sup_{\theta,\theta' \in \Theta} 2L_\Theta \mathbb{E}\left[\|\mathcal{P}_\mathcal{C} - I\|_2\right] \times d(\theta, \theta')\left[\mathbb{E}\left[L^{(\mathcal{C})} + L^{(\mathbb{B}_2(\mathbf{x}))}\right]\right] \\
&= 2L_\Theta \Delta_d(\Theta)\mathbb{E}\left[L^{(\mathcal{C})} + L^{(\mathbb{B}_2(\mathbf{x}))}\right] \mathbb{E}\left[\|\mathcal{P}_\mathcal{C}(\mathbf{x}) - \mathbf{x}\|_2\right].
\end{aligned}
$$

The second equality is obtained by identity $\|\mathbf{a}\|_\mu^2 - \|\mathbf{b}\|_\mu^2 = \langle \mathbf{a} - \mathbf{b}, \mathbf{a} + \mathbf{b}\rangle_\mu$. Then the third inequality follows from the Cauchy-Schwarz inequality. The fourth inequality follows from triangular inequality. The fifth inequality follows from the Lipschitz continuity of $f$ with respect to inputs and parameters. $\qquad\square$

**Corollary 5.** *Under the settings of Lemma 3, if $\mathbf{x}$ is a sub-Gaussian vector with proxy variance $\sigma^2$ and $L^{(\mathcal{A})} = a\sup_{\mathbf{z} \in \mathcal{A}} \|\mathbf{z}\| + b$ for some $a, b \in \mathbb{R}^+$, then there are universal constants $C_1, C_2 > 0$ such that*

$$B_{nrm}(\mathbb{E}[\mathbf{x}] + \mathbb{B}_2(z)) \leq C_1 \sigma^2(\|\mathbb{E}[\mathbf{x}]\| + a\sigma + b)L_\Theta \Delta_d(\Theta)\exp\left(-C_2 z^2/\sigma^2\right). \qquad (26)$$

*Proof.* From Lemma 3, we have

$$B_{nrm}(\mathcal{C}) \leq 2L_\Theta \Delta_d(\Theta)\mathbb{E}\left[L^{(\mathcal{C})} + L^{(\mathbb{B}_2(\mathbf{x}))}\right] \mathbb{E}\left[\|\mathcal{P}_\mathcal{C}(\mathbf{x}) - \mathbf{x}\|_2\right].$$

Now since $\mathbf{x}$ is a sub-Gaussian vector with proxy variance $\sigma^2$, we utilize the results from Lemma 4 and Lemma **??** to obtain the desired result. $\qquad\square$

**Lemma 4.** *Consider the function $f(\mathbf{x}) = a\|\mathbf{x}\| + b$, for some $a, b \in \mathbb{R}^+$. If $\mathbf{x}$ is a sub-Gaussian vector with proxy variance $\sigma^2$, there is a universal constant $C_3 > 0$ such that*

$$\mathbb{E}\left[f(\mathbf{x})\right] \leq C_3 a\sigma + b. \tag{27}$$

*Proof.* By linearity of expectation, we have

$$\mathbb{E}\left[f(\mathbf{x})\right] = a\mathbb{E}\left[\|\mathbf{x}\|\right] + b.$$

For any sub-Gaussian vector $\mathbf{x}$ with proxy variance $\sigma^2$ (see Vershynin (2018)), we have

$$\mathbb{E}\left[\|\mathbf{x}\|\right] \leq C_3\sigma, \tag{28}$$

where $C_3$ is a universal constant. Substituting this into the expectation obtains our desired result. $\square$

**Lemma 5.** *If $\mathbf{x}$ is a sub-Gaussian vector with proxy variance $\sigma^2$, then for any $z > 0$ there are universal constants $C_1, C_2 > 0$ such that*

$$\mathbb{E}\left[\|\mathbf{x} - \mathcal{P}_{\mathbb{E}[\mathbf{x}]+\mathbb{B}_2(z)}(\mathbf{x})\|_2^2\right] \leq C_1\sigma^2\exp\left(-C_2 z^2/\sigma^2\right). \tag{29}$$

*Proof.* By the definition of the projection operator, we have

$$\mathbf{x} - \mathcal{P}_{\mathbb{E}[\mathbf{x}]+\mathbb{B}_2(z)}(\mathbf{x}) = \left[1 - \frac{z}{\|\mathbf{x} - \mathbb{E}[\mathbf{x}]\|_2}\right]_+ (\mathbf{x} - \mathbb{E}[\mathbf{x}]).$$

Taking the expectation of the squared norm, we have

$$\mathbb{E}\left[\|\mathbf{x} - \mathcal{P}_{\mathbb{E}[\mathbf{x}]+\mathbb{B}_2(z)}(\mathbf{x})\|_2^2\right] = \mathbb{E}\left[\left[1 - \frac{z}{\|\mathbf{x} - \mathbb{E}[\mathbf{x}]\|_2}\right]_+^2 \|\mathbf{x} - \mathbb{E}[\mathbf{x}]\|_2^2\right].$$

By the homogeneity of ReLU, we can factor out the squared norm to get

$$\mathbb{E}\left[\|\mathbf{x} - \mathcal{P}_{\mathbb{E}[\mathbf{x}]+\mathbb{B}_2(z)}(\mathbf{x})\|_2^2\right] = \mathbb{E}\left[[\|\mathbf{x} - \mathbb{E}[\mathbf{x}]\|_2 - z]_+^2\right],$$

$$= \int_0^\infty \mathbb{P}\left([\|\mathbf{x} - \mathbb{E}[\mathbf{x}]\|_2 - z]_+^2 > k\right) dk$$

$$= \int_0^\infty \mathbb{P}\left([\|\mathbf{x} - \mathbb{E}[\mathbf{x}]\|_2 - z]_+ > \sqrt{k}\right) dk$$

$$= \int_0^\infty \mathbb{P}\left(\|\mathbf{x} - \mathbb{E}[\mathbf{x}]\|_2 > z + \sqrt{k}\right) dk.$$

For some universal constants $c_1', c_2' > 0$ we have that

$$\mathbb{E}\left[\|\mathbf{x} - \mathcal{P}_{\mathbb{E}[\mathbf{x}]+\mathbb{B}_2(z)}(\mathbf{x})\|_2^2\right] \leq c_1' \int_0^\infty \exp\left(-c_2(z + \sqrt{k})^2/\sigma^2\right) dk.$$

Set $u = z + \sqrt{k}$, then we have $dk = 2(u - z)du$. Substituting this into the integral gives us

$$\mathbb{E}\left[\|\mathbf{x} - \mathcal{P}_{\mathbb{E}[\mathbf{x}]+\mathbb{B}_2(z)}(\mathbf{x})\|_2^2\right] \leq 2c_1' \int_z^\infty (u - z)\exp\left(-c_2 u^2/\sigma^2\right) du,$$

$$= 2c_1' \int_z^\infty u\exp\left(-c_2 u^2/\sigma^2\right) du - 2c_1' z \int_z^\infty \exp\left(-c_2 u^2/\sigma^2\right) du.$$

For the first term, we can use the substitution $v = c_2' u^2/\sigma^2$, which gives us $(\sigma^2/c_2')dv = 2udu$. Observe that second term is negative, so we can upper-bound it by 0. Thus, we have

$$\mathbb{E}\left[\|\mathbf{x} - \mathcal{P}_{\mathbb{E}[\mathbf{x}]+\mathbb{B}_2(z)}(\mathbf{x})\|_2^2\right] \leq \frac{\sigma^2 c_1'}{c_2'} \int_{c_2' z^2/\sigma^2}^\infty \exp(-v)dv = \frac{c_1'}{c_2'}\sigma^2\exp(-c_2' z^2/\sigma^2).$$

By setting $C_1 = c_1'/c_2'$ and $C_2 = c_2'$, we obtain the desired result. $\square$

**Lemma 6.** *Under the assumption 2, the map $\Phi$ satisfies the following statements:*

1. *For any* $\mathbf{H} \in \mathcal{H}$, *and* $\mathbf{x}, \mathbf{x}' \in \mathbb{R}^{n_x}$ *we have that*

$$\|\Phi(\mathbf{H}, \mathbf{x}) - \Phi(\mathbf{H}, \mathbf{x}')\|_2 \leq \left(\prod_{l \in [L]} C_l\right) \left[\sup_{\mathbf{h} \in \mathcal{H}} \|\mathcal{F}_l \mathbf{h}\|_2\right] \|\mathbf{x} - \mathbf{x}'\|_2.$$

2. *For any* $\mathbf{x} \in \mathcal{C}$ *and* $\mathbf{H}, \mathbf{H}' \in \mathcal{H}$, *and some positive constants* $c_1, c_2$ *dependent on* $\{C_l\}$ *we have*

$$\|\Phi(\mathbf{H}, \mathbf{x}) - \Phi(\mathbf{H}', \mathbf{x})\|_2 \leq \left[c_1 \sup_{\mathbf{x} \in \mathcal{C}} \|\mathbf{x}\| + c_2 \sup_{l \in [L], \mathbf{h} \in \mathcal{H}} \|\mathcal{F}_l \mathbf{h}\|_2\right]$$

$$\times \left[\sup_{l \in [L], c \in [C_l], g \in [C_{l-1}]} \|\mathcal{F}_l \mathbf{h}_l^{(c,g)} - \mathcal{F}_l \mathbf{h}_l'^{(c,g)}\|_2\right].$$

*Proof.* Statement 1 is straight forward, each layer and channel has a input Lipschitz constant $\|\sum_k h_{k,l,c,g} S_l^k\|_2 = \|\mathcal{F}_l \mathbf{h}_l^{(c,g)}\|_2$. Therefore, the effective is just layer-wise product of the Lipschitz constants, and using the inequality $\|\mathbf{z}\|_2 \leq \sqrt{d}\|\mathbf{z}\|_\infty$, when $\mathbf{z} \in \mathbb{R}^d$.

- For Statement 2 it is cumbersome, but a useful trick is the identity

$$\|\phi_l(\mathbf{h}, \mathbf{x}) - \phi_l(\mathbf{h}', \mathbf{x}')\|_2 \leq [\|\mathbf{x}\|_2 + \|\mathcal{F}_l \mathbf{h}'\|_2] \max\{\|\mathcal{F}_l \mathbf{h} - \mathcal{F}_l \mathbf{h}'\|_2, \|\mathbf{x} - \mathbf{x}'\|_2\}.$$

- For $c$ channel at layer $l$, we have

$$\|\mathbf{x}_l^c - \mathbf{x}_l'^c\|_2 \leq \sum_{g \in [C_{l-1}]} \left[\|\mathbf{x}_{l-1}^g\|_2 + \|\mathcal{F}_l \mathbf{h}_l'^{(c,g)}\|_2\right] \max\{\|\mathcal{F}_l \mathbf{h}_l^{(c,g)} - \mathcal{F}_l \mathbf{h}_l'^{(c,g)}\|_2, \|\mathbf{x}_{l-1}^g - \mathbf{x}_{l-1}'^g\|_2\}.$$

- At layer $l$ for some positive constants $K_1, K_2$ we have

$$\|\begin{bmatrix} \mathbf{x}_l^1 \\ \vdots \\ \mathbf{x}_l^{C_l} \end{bmatrix} - \begin{bmatrix} \mathbf{x}_l'^1 \\ \vdots \\ \mathbf{x}_l'^{C_l} \end{bmatrix}\|_2 \leq K_1 \left[\|\begin{bmatrix} \mathbf{x}_{l-1}^1 \\ \vdots \\ \mathbf{x}_{l-1}^{C_{l-1}} \end{bmatrix}\|_2 + K_2 \sup \|\mathcal{F}_l \mathbf{h}_l'^{(c,g)}\|_2\right]$$

$$\max\left\{\sup_{c \in [C_l], g \in [C_{l-1}]} \|\mathcal{F}_l \mathbf{h}_l^{(c,g)} - \mathcal{F}_l \mathbf{h}_l'^{(c,g)}\|_2, \|\begin{bmatrix} \mathbf{x}_l^1 \\ \vdots \\ \mathbf{x}_l^{C_{l-1}} \end{bmatrix} - \begin{bmatrix} \mathbf{x}_l'^1 \\ \vdots \\ \mathbf{x}_l'^{C_{l-1}} \end{bmatrix}\|_2\right\}$$

- We recursively apply this inequality for all layers.

$\square$

### A.6   PROOF OF COROLLARY 1

First we state a classical theorem from Shalev-Shwartz et al. (2009) with our notation that we will use in the proof.

**Theorem 3** (Theorem 5 (Shalev-Shwartz et al., 2009)). *Let* $\mathcal{H} \subset \mathbb{R}^d$ *be bounded by* $R$ *and let* $\Phi(\mathbf{H}, \mathbf{z})$ *be* $G$-*Lipschitz continuous with respect to* $\mathbf{H}$ *for any* $\mathbf{z}$. *Then with probability at least* $1 - \delta$ *over a sample of size* $N$, *for all* $\mathbf{H} \in \Theta$ *we have:*

$$\text{GE}(\mathbf{H}) \leq \mathcal{O}\left(GB\sqrt{\frac{d \ln(N) \ln(d/\delta)}{N}}\right) \tag{30}$$

**Proof of Corollary 1.** For single-layer, single-channel GCN under the Assumption 2, we have that $\Phi(\mathbf{H}, \mathbf{x})$ is $\|\mathbf{x}\|_2$-Lipschitz continuos with respective to $\mathbf{H}$. From the boundedness assumption of the data, we have that $\Phi(\mathcal{H}, \mathbf{x})$ is $G$-Lipschitz continuos with $\mathbf{H}$ for all inputs $\mathbf{x}$. Finally since all the filter coefficients are bounded, i.e, $\mathcal{H} \subset \mathbb{R}^{n_f}$ we can instantiate Theorem 3. This concludes our proof.

### A.7 Proofs of Corollary 2

In this section, we provide the proof of Corollary 2. The proof relies on substituting the metric entropy of bounded spectral filters into Theorem 1 and computing the infimum. First we provide classical results on the metric entropy of finite-dimensional spaces which Corollary 2 and 6 are based on.

**Lemma 7** (Volume ratios and metric entropy Wainwright (2019))**.** *Consider a pair of norms* $\|\cdot\|$ *and* $\|\cdot\|'$ *on* $\mathbb{R}^d$, *and let* $\mathbb{B}$ *and* $\mathbb{B}'$ *be the unit balls (i.e.,* $\mathbb{B} := \{\theta \in \mathbb{R}^d \mid \|\theta\| \leq 1\}$, *with* $\mathbb{B}'$ *similarly defined). Then the* $\varepsilon$-*covering number of* $\mathbb{B}$ *in the norm* $\|\cdot\|'$ *obeys the bounds*

$$\left(\frac{1}{\varepsilon}\right)^d \frac{\mathsf{vol}(\mathbb{B})}{\mathsf{vol}(\mathbb{B}')} \leq \mathcal{N}(\mathbb{B}, \|\cdot\|', \varepsilon) \leq \left(1 + \frac{2}{\varepsilon}\right)^d \frac{\mathsf{vol}(\mathbb{B})}{\mathsf{vol}(\mathbb{B}')}. \tag{31}$$

*As a special case, if* $\|\cdot\|$ *and* $\|\cdot\|'$ *are the same norm, then the metric entropy satisfies the bound*

$$d\ln(1/\varepsilon) \leq \ln(\mathcal{N}(\mathbb{B}, \|\cdot\|, \varepsilon)) \leq d\ln(1 + 2/\varepsilon). \tag{32}$$

Lemma 7 gives us a way to compute the metric entropy for the scenario in Corollary 2. Next, we introduce an auxilary proposition that helps us compute the inf in Theorem 1.

**Proposition 2.** *Suppose* $a > 0$ *then the following holds true:*

$$\inf_{x>0} \left(x + a\sqrt{\ln(1 + 1/x)}\right) < 2a \cdot \max\{1, \sqrt{\ln(1 + 1/a)}\} \tag{33}$$

*Proof.* Define $f(x, a) := \left(x + a\sqrt{\ln(1 + 1/x)}\right)$, clearly by the definition of infimum we have $\inf_{x>0} f(x, a) < f(x, a)$. Choose $x = a$, then we have

$$\inf_{x>0} f(x, a) < a\left(1 + \sqrt{\ln(1 + 1/a)}\right), \tag{34}$$

when $a \geq 1/(e-1)$ we have $\ln(1 + 1/a) \leq 1$, therefore we can upper bound $\inf_x f(x, a) < 2a$. When $a < 1/(e-1)$ we have $\ln(1 + 1/a) > 1$, therefore we can upper bound

$$\inf_x f(x, a) < a\sqrt{\ln(1 + 1/a)} + a\sqrt{\ln(1 + 1/a\sqrt{\ln(1 + 1/a)})} \leq 2a\sqrt{\ln(1 + 1/a)}. \tag{35}$$

This concludes the proof. $\square$

**Proof of Corollary 2.** We instantiate Theorem 1 $C_l = 1$, with metric entropy from Lemma 7, we have

$$\mathbb{P}\left(\mathrm{GE}(\hat{\mathbf{H}}_N) > \inf_{\varepsilon \in (0,K]} \left\{2\varepsilon + K\sqrt{\frac{Ln_x \ln(1 + 2\max\{K', K+1\}/\varepsilon) + \ln(3/\delta)}{2N}}\right\}\right) \leq \delta.$$

Define $K'' = \max\{K', K+1\}$. Now re-scale $\varepsilon = 2K''\varepsilon'$ this gives us

$$\mathbb{P}\left(\mathrm{GE}(\hat{\mathbf{H}}_N) > \inf_{\varepsilon' \in (0,K/2K'']} \left\{4K''\varepsilon' + K\sqrt{\frac{Ln_x \ln(1 + 1/\varepsilon') + \ln(3/\delta)}{2N}}\right\}\right) \leq \delta.$$

We take $4K''$ as common to have

$$\mathbb{P}\left(\mathrm{GE}(\hat{\mathbf{H}}_N) > 4K'' \cdot \inf_{\varepsilon' \in (0,K/2K'']} \left\{\varepsilon' + \frac{K}{4K''\sqrt{2N}}\sqrt{\frac{Ln_x \ln(1 + 1/\varepsilon') + \ln(3/\delta)}{2N}}\right\}\right) \leq \delta.$$

Now we upper bound the summands in the square root to have

$$\mathbb{P}\left(\mathrm{GE}(\hat{\mathbf{H}}_N) > K\sqrt{\frac{\ln(3/\delta)}{2N}} + 4K'' \cdot \inf_{\varepsilon' \in (0,K/2K'']} \left\{\varepsilon' + \frac{K}{4K''}\sqrt{\frac{Ln_x}{2N}}\sqrt{\ln(1 + 1/\varepsilon')}\right\}\right) \leq \delta.$$

We upper bound the inf using the Proposition 2 to have

$$\mathbb{P}\left(\mathrm{GE}(\hat{\mathbf{H}}_N) > K\sqrt{\frac{\ln(3/\delta)}{2N}} + K\sqrt{\frac{2Ln_x \ln\left(1 + \max\left\{\frac{4K''}{K}\sqrt{\frac{2N}{Ln_x}}, e-1\right\}\right)}{N}}\right) \leq \delta.$$

This concludes the proof.

### A.8 PROOFS OF COROLLARY 6

Authors of Yang et al. (2022) observed that GCN filter spectra are not only bounded but also sparse on real-world datasets like ZINC (Irwin et al., 2012), suggesting that Corollary 2 can be tightened. We now present generalization bounds for $s$-sparse spectra, i.e., $\|\mathsf{diag}^\dagger(\tilde{h}(\Lambda))\|_0 \le s$.

**Corollary 6.** *Let various symbols be as in Theorem 1 with $C_l = 1$. If $\mathcal{FH}$ is bounded by 1, and $s$-sparse, then w.p. of at least $1 - \delta$ we have*

$$\mathrm{GE}\left(\hat{\mathbf{H}}_N\right) \le K\sqrt{\frac{2Ls}{N}\ln\left(1 + \max\left\{\frac{4K''}{K}\sqrt{\frac{2N}{sL}}, e - 1\right\}\right)} + K\sqrt{\frac{s\ln\left(en_x/s\right) + \ln\left(\frac{3}{\delta}\right)}{2N}}. \tag{36}$$

**Remarks.** The sample complexity is improved to $N \ge \mathcal{O}\left(Ls + s\ln(en_x/s) + \ln(1/\delta)\right)$ in comparison to Corollary 2's $N \ge \mathcal{O}\left(Ln_x + \ln(1/\delta)\right)$.

*Proof.* We use the classical result on the covering numbers for $s$-sparse vectors (Chandrasekaran et al., 2012), the metric entropy satisfies $\ln(\mathcal{FH}, \|\cdot\|_2, \varepsilon) \le \mathcal{O}(s\ln(1/\varepsilon) + s\ln(en_x/s))$. The rest follows as in Corollary 2. Now we state the metric entropy of the set of sparse vectors.

**Lemma 8** (Metric entropy of sparse vectors). *The $\varepsilon$-metric entropy of the set of $s$-sparse vectors $\mathbb{B}_s = \left\{\theta \in \mathbb{R}^d : \|\theta\|_0 \le s \cap \|\theta\|_2 \le 1\right\}$ is given by*

$$\ln(\mathcal{N}(\mathbb{B}_s, \|\cdot\|_2, \varepsilon)) \le s\ln(1 + 2/\varepsilon) + s\ln(en_x/s). \tag{37}$$

*Proof.* For a fixed support $S \subseteq [d]$ of size $s$, the set of vectors is a $s$-dimensional ball in $\mathbb{R}^s$ with radius 1. Lemma 7 gives us the upper bound $(1 + 2/\varepsilon)^s$. Since we do not know the support, we need to choose $s$ coordinates from $d$ and then cover the ball. The number of ways to choose $s$ coordinates is $\binom{d}{s} \le (ed/s)^s$. Therefore, the effective metric entropy is

$$\ln(\mathcal{N}(\mathbb{B}_s, \|\cdot\|_2, \varepsilon)) \le s\ln(1 + 2/\varepsilon) + s\ln(ed/s).$$

$\square$

**Proof of Corollary 6.** Following the proof technique of Corollary 2, define $K'' = \max\{K', K + 1\}$, we have

$$\mathbb{P}\left(\mathrm{GE}(\hat{\mathbf{H}}_N) > \inf_{\varepsilon \in (0, K]}\left\{2\varepsilon + K\sqrt{\frac{Ls\ln(1 + 2K''/\varepsilon) + Ls\ln(en_x/s) + \ln(3/\delta)}{2N}}\right\}\right) < \delta.$$

Now we re-scale $\varepsilon = 2K''\varepsilon'$, this gives us

$$\mathbb{P}\left(\mathrm{GE}(\hat{\mathbf{H}}_N) > \inf_{\varepsilon' \in (0, K/2K'']}\left\{4K''\varepsilon' + K\sqrt{\frac{Ls\ln(1 + 1/\varepsilon') + Ls\ln(en_x/s) + \ln(3/\delta)}{2N}}\right\}\right) < \delta.$$

By upper bounding the summands in the square root we have

$$\mathbb{P}\left(\mathrm{GE}(\hat{\mathbf{H}}_N) > K\sqrt{\frac{Ls\ln(en_x/s) + \ln(3/\delta)}{2N}} + \inf_{\varepsilon' \in (0, K/2K'']}\left\{4K''\varepsilon' + K\sqrt{\frac{Ls\ln(1 + 1/\varepsilon')}{2N}}\right\}\right)$$
$$< \delta.$$

We take $4K''$ as common to have

$$\mathbb{P}\left(\mathrm{GE}(\hat{\mathbf{H}}_N) > K\sqrt{\frac{Ls\ln(en_x/s) + \ln(3/\delta)}{2N}}\right.$$
$$\left. + 4K'' \cdot \inf_{\varepsilon' \in (0, K/2K'']}\left\{\varepsilon' + \frac{K}{4K''}\sqrt{\frac{Ls}{2N}}\sqrt{\ln(1 + 1/\varepsilon')}\right\}\right) < \delta.$$

We upper bound the inf using Proposition 2 to have

$$\mathbb{P}\left(\mathrm{GE}(\hat{\mathbf{H}}_N) > K\sqrt{\frac{Ls\ln(en_x/s) + \ln(3/\delta)}{2N}} + K\sqrt{\frac{2Ls}{N}\ln\left(1 + \max\left\{\frac{4K''}{K}\sqrt{\frac{2N}{s}}, e - 1\right\}\right)}\right)$$

$$< \delta.$$

This concludes the proof. □

### A.9 Proofs of Corollary 3

Next, we move onto the metric entropy of infinite-dimensional spaces. First, we consider the set of all Lipschitz functions on the $d$-dimensional unit ball $\mathbb{B}^d$.

**Lemma 9** (Metric entropy of Lipschitz functions Wainwright (2019)). *Consider the class of Lipschitz functions*

$$\mathscr{F} := \left\{f : [0,1]^d \to \mathbb{R} \mid f(0) = 0, \text{ and } \|f(\mathbf{x}) - f(\mathbf{x}')\|_2 \le L\|\mathbf{x} - \mathbf{x}'\|_2 \quad \forall \mathbf{x}, \mathbf{x}' \in [0,1]^d\right\}.$$

*Then the $\varepsilon$-metric entropy of $\mathscr{F}$ on the sup-norm $\|\cdot\|_\infty$ satisfies*

$$\ln(\mathcal{N}(\mathscr{F}, \|\cdot\|_\infty, \varepsilon)) \le \mathcal{O}\left((L/\varepsilon)^d\right). \tag{38}$$

**Lemma 10.** *Consider the set $\mathcal{A}(\{\lambda_j\}) := \{\{x_i\} : \forall i \in \mathbb{N}, f \in \mathcal{A}; x_i = h(\lambda_i)\}$, where $\mathcal{A} := \{f : \mathbb{C} \to \mathbb{C} : \forall \lambda, \lambda' \in \mathbb{C}, |f(\lambda) - f(\lambda')| \le L|\lambda - \lambda'|, \|f\|_\infty \le 1, f(0) = 0\}$. Then for any $\varepsilon > 0$ the metric entropy satisfies the inequality:*

$$\ln(\mathcal{N}(\mathcal{A}(\{\lambda_j\}), \|\cdot\|_2, \varepsilon)) \le \mathcal{O}\left((L/\varepsilon)^4\right). \tag{39}$$

*Proof.* Consider two elements $\{x_i\}, \{x_i'\} \in \mathcal{A}(\{\lambda_j\})$, then we have $\|\{x_i\} - \{x_i'\}\|_\infty = \|\{h(\lambda_i)\} - \{h'(\lambda_i)\}\|_\infty$ for some $h, h' \in \mathcal{A}$. By definition we have that

$$\|\{x_i\} - \{x_i'\}\|_\infty = \|\{h(\lambda_i)\} - \{h'(\lambda_i)\}\|_\infty \le \|h - h'\|_\infty. \tag{40}$$

For any arbitrary $\{x_i\} \in \mathcal{A}(\{\lambda_j\})$ generated by $h \in \mathcal{A}$, we can find a function $h' \in \mathcal{A}$ such that $\|h - h'\|_\infty \le \varepsilon$ generating a sequence $\{h'(\lambda_i)\}$. Now clearly, $\varepsilon$-net of $\mathcal{A}$ can generate a sequence that forms a $\varepsilon$-net of $\mathcal{A}(\{\lambda_j\})$.

Therefore, we have that $\ln(\mathcal{N}(\mathcal{A}(\{\lambda_j\}), \|\cdot\|_\infty, \varepsilon)) \le \ln(\mathcal{N}(\mathcal{A}, \|\cdot\|_\infty, \varepsilon))$. We obtain the desired inequality by invoking result from Lemma 9 using the fact that $\mathbb{C} \cong \mathbb{R}^2$. □

Next, we provide an auxiliary proposition that helps us compute the infimum in Theorem 1 for the scenario in Corollary 3.

**Proposition 3.** *Suppose $a, p > 0$ then the following holds true:*

$$\inf_{x>0}\left(x + ax^{-p}\right) < \left(p^{1/(1+p)} + p^{-p/(1+p)}\right)a^{1/(1+p)}. \tag{41}$$

*Proof.* We set the derivative of the objective to zero, i.e,

$$1 - apx^{-p-1} = 0 \Rightarrow x^\star = (ap)^{1/(p+1)}.$$

Plugging this into the objective gives us the desired result. □

**Proof of Corollary 3.** Now we instantiate Theorem 1 with $C_l = 1$, and metric entropy from Lemma 10. Then for some constant $C > 0$ we have

$$\mathbb{P}\left(\mathrm{GE}(\hat{\mathbf{H}}_N) > \inf_{\varepsilon \in (0,K]}\left\{2\varepsilon + K\sqrt{\frac{CL(P/\varepsilon)^4 + \ln(3/\delta)}{2N}}\right\}\right) \le \delta,$$

where $K'' = \max\{K', K+1\}$. We re-scale $\varepsilon = P\varepsilon'/C^{1/4}$, this gives us

$$\mathbb{P}\left(\mathrm{GE}(\hat{\mathbf{H}}_N) > \inf_{\varepsilon' \in (0, KC^{1/4}/P]}\left\{\frac{2P}{C^{1/4}}\varepsilon' + K\sqrt{\frac{L/\varepsilon'^4 + \ln(3/\delta)}{2N}}\right\}\right) \le \delta.$$

Now we upper bound the summands in the square root to have

$$\mathbb{P}\left(\text{GE}(\hat{\mathbf{H}}_N) > K\sqrt{\frac{\ln(3/\delta)}{2N}} + \frac{2P}{C^{1/4}} \cdot \inf_{\varepsilon' \in (0, KC^{1/4}/P]} \left\{\varepsilon' + \frac{KC^{1/4}}{P\sqrt{8N/L}}\varepsilon'^{-2}\right\}\right) \leq \delta.$$

From the Proposition 2 with $p = 2$, we have

$$\mathbb{P}\left(\text{GE}(\hat{\mathbf{H}}_N) > K\sqrt{\frac{\ln(3/\delta)}{2N}} + \frac{2P}{C^{1/4}}\left(2^{1/3} + 2^{-2/3}\right)\left(\frac{KC^{1/4}}{P\sqrt{8N/L}}\right)^{1/3}\right) \leq \delta.$$

Simplifying the constants gives us the desired result.

## A.10 Proofs of Corollary 4

**Lemma 11.** *Consider the set of countably infinite dimensional vector whose entries polynomially decay as $C/i^\alpha$, i.e., $\mathcal{A} := \left\{\{x_i\} \in (\mathbb{R}) : \forall i \in \mathbb{N}; |x_i| \leq \frac{C}{i^\alpha}\right\}$. If $\alpha > 1/2$, then for any $\varepsilon > 0$ the metric entropy satisfies the inequality:*

$$\ln(\mathcal{N}(\mathcal{A}, \|\cdot\|_2, \varepsilon)) \leq \left(\frac{16C^2}{2\alpha - 1}\right)^{1/(2\alpha-1)} \varepsilon^{-2/(2\alpha-1)} \ln\left(1 + \frac{4C\sqrt{\zeta(2\alpha)}}{\varepsilon}\right), \qquad (42)$$

*where $\zeta(\cdot)$ is the Riemann zeta function.*

*Proof.* We will construct a sub-set defined as

$$\mathcal{A}_d := \left\{\{y_i\} \in (\mathbb{R}) : \forall i \in [d], |y_i| \leq \frac{C}{i^\alpha} \text{ and } \forall j > d, y_j = 0\right\}. \qquad (43)$$

For any element $\{x_i\}, \{x_i'\} \in \mathcal{A}, \{y_i\} \in \mathcal{A}_d$ and $\{y_i'\} \in \mathcal{C}(\mathcal{A}_d, \varepsilon/2)$ we have the inequality

$$\|\{x_i\} - \{x_i'\}\|_2 \leq \|\{x_i\} - \{y_i\}\|_2 + \|\{y_i'\} - \{x_i'\}\|_2 + \|\{y_i\} - \{y_i'\}\|_2. \qquad (44)$$

Using the fact that $\{y_i'\} \in \mathcal{C}(\mathcal{A}_d, \varepsilon/2)$ we have

$$\|\{x_i\} - \{x_i'\}\|_2 \leq \varepsilon/2 + \|\{x_i\} - \{y_i\}\|_2 + \|\{y_i'\} - \{x_i'\}\|_2, \qquad (45)$$

we now apply the triangular inequality to obtain,

$$\|\{x_i\} - \{x_i'\}\|_2 \leq \varepsilon/2 + 2\sqrt{\sum_{j>d}\left(\frac{C}{j^\alpha}\right)^2} \leq \frac{\varepsilon}{2} + \frac{2C}{\sqrt{2\alpha - 1}}d^{-(2\alpha-1)/2}. \qquad (46)$$

Now choose $d$ such that $\frac{2C}{\sqrt{2\alpha-1}}d^{-(2\alpha-1)/2} \leq \varepsilon/2$. This requires that

$$d \geq d(\varepsilon) := \left(\frac{4C}{\sqrt{2\alpha - 1}}\varepsilon^{-1}\right)^{2/(2\alpha-1)}.$$

When $d \geq d(\epsilon)$ we have that $\|\{x_i\} - \{x_i'\}\|_2 \leq \varepsilon$. By definition the metric entropy of $\mathcal{A}$ is smaller than the metric entropy of $\mathcal{A}_d(\varepsilon)$. Then

$$\ln(\mathcal{N}(\mathcal{A}, \|\cdot\|_2, \varepsilon)) \leq \ln(\mathcal{N}(\mathcal{A}_{d(\varepsilon)}, \|\cdot\|_2, \varepsilon/2)) \leq d(\varepsilon)\ln\left(1 + \frac{4}{\varepsilon}\sqrt{\left(\sum_{i>0} C^2/i^{2\alpha}\right)}\right). \qquad (47)$$

On simplification we have

$$\ln(\mathcal{N}(\mathcal{A}, \|\cdot\|_2, \varepsilon)) \leq \left(\frac{16C^2}{2\alpha - 1}\right)^{1/(2\alpha-1)} \varepsilon^{-2/(2\alpha-1)} \ln\left(1 + \frac{4C\sqrt{\zeta(2\alpha)}}{\varepsilon}\right). \qquad (48)$$

This gives us the desired result. $\qquad\square$

**Corollary 7.** *Consider the set of countably infinite dimensional vector whose entries polynomially decay as $C/i^\alpha$, i.e., $\mathcal{A} := \left\{ \{x_i\} \in (\mathbb{C}) : \forall i \in \mathbb{N}; |x_i| \le \frac{C}{i^\alpha} \right\}$. If $\alpha > 1/2$, then for any $\varepsilon > 0$ the metric entropy satisfies the inequality:*

$$\ln(\mathcal{N}(\mathcal{A}, \|\cdot\|_2, \varepsilon)) \le 2 \left( \frac{16 C^2}{2\alpha - 1} \right)^{1/(2\alpha - 1)} \varepsilon^{-2/(2\alpha-1)} \ln\left( 1 + \frac{4C\sqrt{\zeta(2\alpha)}}{\varepsilon} \right), \qquad (49)$$

*where $\zeta(\cdot)$ is the Riemann zeta function.*

*Proof.* The proof is same as Lemma 11, except that the metric entropy of $\mathcal{A}_{d(\varepsilon)}$ is upper bounded by $2d(\varepsilon) \ln\left( 1 + \frac{4}{\varepsilon}\sqrt{\left( \sum_{i>0} C^2/i^{2\alpha} \right)} \right)$. $\qquad \square$

**Proposition 4.** *Suppose $a, p > 0$ then the following holds true:*

$$\inf_{x>0} \left( x + a\sqrt{x^{-2p} \ln(1 + 1/x)} \right) < 2(ap)^{1/(p+1)} \sqrt{\ln\left( 1 + \max\{(ap)^{-1/(1+p)}, e^{p^2} - 1\} \right)}. \quad (50)$$

*Proof.* Choose $x = (ap)^{1/(p+1)}$ as the candidate point to upper bound the infimum. Then we have

$$\inf_{x>0} \left( x + a\sqrt{x^{-2p} \ln(1 + 1/x)} \right) < (ap)^{1/(p+1)} + a\sqrt{(ap)^{-2p/(p+1)} \ln(1 + (ap)^{-1/(p+1)})}.$$

Taking $(ap)^{1/(p+1)}$ out common we have

$$\inf_{x>0} \left( x + a\sqrt{x^{-2p} \ln(1 + 1/x)} \right) < (ap)^{1/(p+1)} \left( p + \sqrt{\ln(1 + (ap)^{-1/(p+1)})} \right).$$

By applying Cauchy-Schwarz inequality we have the desired result. $\qquad \square$

**Proof of Corollary 4.** From Theorem 1 and Corollary 7 we have

$$\mathbb{P}\left( \text{GE}(\hat{\mathbf{H}}_N) > \inf_{\varepsilon \in (0,K]} \left\{ 2\varepsilon + K\sqrt{\frac{w_1 L \varepsilon^{-2/(2k-1)} \ln(1 + w_2/\varepsilon) + \ln(3/\delta)}{2N}} \right\} \right) < \delta,$$

where $w_1 = 2\left( \frac{16 A_{\text{pass}}^2}{2k-1} \right)^{1/(2k-1)}$, and $w_2 = 4 A_{\text{pass}}\sqrt{\zeta(2k)} \cdot \max\{K', K+1\}$.

Now re-scale $\varepsilon = w_2 \varepsilon'$, this gives us

$$\mathbb{P}\left( \text{GE}(\hat{\mathbf{H}}_N) > \inf_{\varepsilon' \in (0, K/w_2]} \left\{ 2w_2\varepsilon' + K\sqrt{\frac{w_1 w_2^{-2/(2k-1)} L \varepsilon'^{-2/(2k-1)} \ln(1 + 1/\varepsilon') + \ln(3/\delta)}{2N}} \right\} \right)$$
$$\le \delta.$$

Now we upper bound the summands in the square root to have

$$\mathbb{P}\left( \text{GE}(\hat{\mathbf{H}}_N) > K\sqrt{\frac{\ln(3/\delta)}{2N}} \right.$$
$$\left. + 2w_2 \cdot \inf_{\varepsilon' \in (0, K/w_2]} \left\{ \varepsilon' + \frac{K}{2w_2^{2k/(2k-1)}} \sqrt{\frac{w_1 L}{2N}} \sqrt{\varepsilon'^{-2/(2k-1)} \ln(1 + 1/\varepsilon')} \right\} \right) \le \delta.$$

From Proposition 4 with $p = 1/(2k-1)$, we have

$$\mathbb{P}\left( \text{GE}(\hat{\mathbf{H}}_N) > K\sqrt{\frac{\ln(3/\delta)}{2N}} + 4\left( \frac{K}{2(2k-1)}\sqrt{\frac{w_1 L}{2}} \right)^{(2k-1)/(2k)} N^{-(2k-1)/4k} \right.$$
$$\left. \times \sqrt{\ln\left( 1 + \max\left\{ w_2\left( \frac{2(2k-1)}{K}\sqrt{\frac{2}{w_1 L}} \right)^{1-1/2k} N^{(2k-1)/4k}, e^{1/(2k-1)^2} - 1 \right\} \right)} \right)$$
$$\le \delta.$$

Now let us apply the limit $k \to \infty$, the term $\lim_{k\to\infty} N^{(2k-1)/4k} \to \sqrt{N}$, $\lim_{k\to\infty} e^{1/(2k-1)^2} - 1 \to 0$. $\lim_{k\to\infty} w_2 \to 4A_{\text{pass}}$ because $\lim_{k\to\infty} \zeta(2k) \to 1$. Finally we have the coefficient term

$$\lim_{k\to\infty} \left( \frac{K}{2(2k-1)} \sqrt{\frac{w_1 L}{2}} \right)^{(2k-1)/(2k)} = \lim_{k\to\infty} \left( \frac{K}{2(2k-1)\sqrt{2}} \right)^{(2k-1)/(2k)} (Lw_1)^{(2k-1)/4k}$$

$$= \lim_{k'\to\infty} \left( \frac{K}{2\sqrt{2}k'} \right)^{1/(k'+1)}$$

$$\times \lim_{k\to\infty} (2L)^{(2k-1)/4k} \left( \frac{16A_{\text{pass}}^2}{2k-1} \right)^{1/4k} \to \sqrt{2L}.$$

Thus when $k \to \infty$ we have

$$\mathbb{P}\left( \text{GE}(\hat{\mathbf{H}}_N) > K\sqrt{\frac{\ln(3/\delta)}{2N}} + 8\sqrt{\frac{L\ln(1 + A_{\text{pass}}\sqrt{8N/L})}{2N}} \right) \le \delta$$

This concludes the proof.