# OpenReview forum: "A Spectral Characterization of Generalization in GCN: Escaping the Curse of Dimensionality"
_ICLR.cc/2026/Conference — Submitted to ICLR 2026_

### Official Review · Reviewer_ggKt · 2025-10-27

**Soundness:** 3
**Presentation:** 3
**Contribution:** 3
**Rating:** 6
**Confidence:** 2

**Summary:**

This paper provides a new theoretical framework for analyzing the generalization properties of GCNs by leveraging classical tools from signal processing and modern statistical learning techniques. The approach differs from prior work in that it leverages the graph spectral structure rather than relying purely on the parameter space. The authors argue that GCN filters, viewed in the graph Fourier domain, have lower intrinsic dimensionality, allowing the derivation of tighter generalization error bounds that are independent of the number of parameters. The paper further extends the analysis to graphons. Numerical experiments validate the scaling of the theoretical bounds.

**Strengths:**

1. Sharp generalization bounds.
2. The paper is generally well-written and clearly motivated.
3. The paper provides some insights in Sec. 4.
4. This seems to be the first theoretical results that analyze generalization properties in the spectral domain. The spectral viewpoint provides an elegant and under-explored angle for bounding generalization in GCNs.
5. Tighter bounds: The derived bounds scale as $\sqrt{n_x/N},$ independent of the total number of parameters, and improved over VC-dimension/PAC-Bayes/Rademacher complexity bounds.

**Weaknesses:**

1. The paper claims sharp bounds but has not shown the actual gap between empirical and theoretical error bounds.
2. The sub-Gaussian assumption may not hold for real-world graph data.
3. The bounds involve constants such as $K, K'', L_\mathcal{X},$ and $L_\mathcal{H},$ which can benefit from better explanation and intuition of why they are defined in this way.
4. Some typos/grammar issues, e.g., in lines 135-136 it should be "an empirical"; in lines 293-294 "While" should be removed.

**Questions:**

1. How tight are your theoretical bounds to empirical values?
2. How sensitive are your results to the Lipschitz or sub-Gaussian assumptions? Would small violations lead to large deviations in generalization behavior?

---

> ### Author Response · Authors · 2025-11-21
>
> We thank the reviewer for their time and the constructive feedback. Below are our detailed responses to each of the weaknesses and questions raised by the reviewer.
>
> ## Responses to Weakness points
>
> 1. As we discussed in our global response, the primary goal of our experiments is to demonstrate the trends in generalization predicted by our theoretical analysis. For the real-world datasets used in our experiments, true constants such as $L_g$, and $\sigma_e$ are unknown. Therefore, it is not possible to compute the exact numerical theoretical bounds. However, to address this concern, we have designed a synthetic experiment with Erdos-Rényi-Gilbert model (or Stochastic Block Model). We now have a comparison between empirical generalization error and theoretical bounds in the updated version (Figure 3). As a sanity check, the upper bounds are always greater than or equal to the empirical generalization error. Although, there is a constant gap between empirical and theoretical values, the trends closely follow each other as we vary the number of training samples $N$. The primary objective of our work is to track the non-asymptotic trends with respective to the problem parameters (like $N, n_x, L$).
>
> 2. Please see our global reply where we address this issue in detail. In brief, however, we respectfully disagree that this is a restrictive assumption and note that this largely relaxes common assumptions made in prior work (which, for example, required data to be bounded). Moreover, all of the datasets that we included in our experiments are either categorical or were verified to have quadratic tails (see Table 2, Appendix A.2), ensuring they are, in fact, sub-Gaussian.
>
> 3. $L_{\mathcal{X}} $ (or $L_{\mathcal{H}} $) is the Lipschitz constant of the GCN w.r.t. the input signal (or parameters). $K$ is the sub-Gaussian proxy variance (or upper bound thereof) of the GCN prediction. $K''$ is just an intermediate constant that appears while applying the uniform concentration techniques. We have spelled out these in detail in the updated version.
>
> 4. We thank the reviewer for pointing out these typos. We have corrected them in the revised version.
>
> ## Response to Questions
>
> 1. Please see our response to Weakness 1 and 2.
>
> 2. The below is our response to both assumptions:
>
>    a) **Sub-Gaussian distribution**: We refer the reviewer to our response to Weakness 2.
>
>    b) **Lipschitz continuity**: We are unsure if the reviewer is referring to the Lipschitz continuity of activation function or the spectrum. Therefore, we respond to both cases.
>       - **Activations:** As discussed in Line 240-241, most commonly used activation functions such as Softmax, ReLU, Leaky ReLU, Tanh, ELU, SELU, and SiLU are all Lipschitz continuous. Activation functions that are not Lipschitz continuous are non-standard and could lead fundamental problems such as unstable training; thereby we do not consider them in our analysis. Nevertheless, in Figure 1(b) we empirically validate the effect of varying Lipschitz constant up-to $10^7$ and we observe that the generalization error saturates after a certain value.
>
>       - **Spectrum:** Violation of local Lipschitz continuity of GCN can only occur in the following scenarios:
>         - *Unbounded spectrum*: This happens when we have a pole in the filter response within the spectrum range. However, this is not possible for polynomial filters.
>         - *Unbounded eigenvalues*: In practice, the common choice of graph shift operators are normalized adjacency or Laplacian matrices which have bounded eigenvalues.
>         - *Discontinuous spectrum*: Perfect low-pass filter ($k \to \infty$ in Equation (11)) is one such case, and we have covered this case in Corollary 3.
>
>         We believe the first two scenarios are not practical, and the last is covered in our analysis.

---

> > ### Comment · Reviewer_ggKt · 2025-11-21
> >
> > Thank you for your response. I think the paper will benefit from adding those clarifications.

---

> > > ### Author Response · Authors · 2025-11-25
> > >
> > > We thank the reviewer for going through our response. We have incorporated all the clarifications in the updated version of the paper. We hope the reviewer finds the revised version satisfactory. We are happy to further discuss if there are any remaining concerns.

---

### Official Review · Reviewer_8PWR · 2025-10-31

**Soundness:** 2
**Presentation:** 2
**Contribution:** 2
**Rating:** 4
**Confidence:** 4

**Summary:**

The paper presents a theoretical study of generalization in Graph Convolutional Networks (GCNs) from a spectral perspective. It argues that the spectral representation of graph filters provides a natural space to analyze generalization, yielding bounds independent of the number of parameters. The authors derive sharp non-asymptotic generalization bounds that scale nearly linearly with the number of graph nodes and remain finite in the infinite-node (graphon) limit under mild regularity assumptions. Theoretical results are complemented by numerical simulations on both homophilous and heterophilous datasets, verifying the dependence of generalization on Lipschitz constants and spectral properties.

**Strengths:**

1. The spectral characterization of GCN generalization is interesting and deserves in-depth theoretical analysis;
2. The paper includes comparisons with classical frameworks (VC dimension, PAC-Bayes, Rademacher complexity) and state-of-the-art works, highlighting the improvement in sample complexity.

**Weaknesses:**

1. The readability is not high, as the notation is complex and difficult to follow.

2. The link between the simulation studies and the theoretical insights is not clearly established.

3. If the authors focus on the graph classification setup, what about the case of node classification?

4. The current theoretical result does not consider the over-smoothing issue, why?

**Questions:**

1. How sensitive are the derived bounds to violations of the sub-Gaussian assumption?

2. Can the spectral regularity (Lipschitz or low-pass) conditions be empirically estimated from real GCNs during training?

3. Would incorporating stochasticity in the graph structure (e.g., random graph models) affect the generalization scaling behavior?

4. Could the theoretical insights guide the design of regularization terms or spectral constraints to improve practical generalization?

---

> ### Author Response · Authors · 2025-11-21
>
> We thank the reviewer for their time and the constructive feedback. Below are our detailed responses to each of the weaknesses and questions raised by the reviewer.
>
> ## Response to Weaknesses
>
> 1. Reviewers ggKt and LuGU have expressed that our draft is well-written. We would appreciate if the reviewer could point out specific sections or notations that are difficult to follow so that we can improve them in the revised version.
>
> 2. We respond to both points 2 and 3 together. In our theoretical framework, we consider a general target vector (Line 101-102), which can represent either node-level outputs (for node classification) or graph-level outputs (for graph classification). The generalization bounds we derived are applicable to both settings as long as the regularity conditions on the graph filters are satisfied. In our experiments, we focused on transductive node classification tasks, that is, each data point corresponds to feature vector of a node and the target is the label of that node which a special case of our modelling, i.e., we can view the inputs in transductive node classification as i.i.d. masked node features. Therefore, we believe that there is no discrepancy between theory and experiments.
>
> 4. We believe the reviewer is referring to the over-smoothing phenomenon in deep GCNs (as in Rusch et. al., '23), where the node features become indistinguishable as the number of layers increases. We argue that over-smoothing is a separate issue from generalization bounds. Specifically, over-smoothing affects the expressivity of the GCN, i.e., its ability to learn meaningful representations from the data. In contrast, our work focuses on providing probabilistic guarantees on how well the learned model generalizes from training data to unseen data, given a fixed architecture and set of parameters. If anything, over-smoothing will improve generalization as it reduces the model complexity.
>
> ## Response to Questions
>
> 1. Extending our theoretical framework to heavy-tailed distributions would lead to slower rates than the current $\mathcal{O}(\sqrt{n_x/N})$ rate as it requires different concentration inequalities (as stated in Line 296 -- 298). However, we believe that this setting is not practical and purely a theoretical adventure because we are unaware of real-world graph datasets with such tail properties. Please refer to the global response for further discussion.
>
> 2. Yes, these quantities can be estimated empirically. Our Lemma 6 in the Appendix provides an upper bound.
>
> 3. Our current theoretical framework assumes that the graph structure is fixed and known a priori, that is, the graph Fourier basis is known. Incorporating stochasticity in the graph structure would break this assumption. This requires identifying a suitable notion of spectral complexity that may arise from the distribution of graphs. This is a non-trivial extension and is left as future work.
>
> 4. Yes, our theoretical insights suggest that controlling the spectral complexity of GCN filters can lead to improved generalization; we have demonstrated this empirically in Section 4. Practically one could use the Lipschitz constant of the filter (obtained from Lemma 6) or spectral values above certain frequency as regularizers during training to encourage better generalization.

---

> > ### Comment · Reviewer_8PWR · 2025-11-25
> >
> > Thanks to the authors for providing the requested clarifications. I am now in favor of accepting this work and encourage the authors to incorporate these clarifications into the revised version of the manuscript for the benefit of future readers.

---

> > > ### Author Response · Authors · 2025-11-25
> > >
> > > We thank the reviewer for going through our response and reconsidering their rating.
> > > We have incorporated all the clarifications in the updated version of the paper.

---

> > > > ### Comment · Reviewer_8PWR · 2025-11-26
> > > >
> > > > If possible, I recommend that the authors provide the code necessary to reproduce the experimental results presented in the paper. This would facilitate future research and support those who may wish to build upon this work.

---

### Official Review · Reviewer_8YBy · 2025-10-31

**Soundness:** 3
**Presentation:** 3
**Contribution:** 2
**Rating:** 4
**Confidence:** 3

**Summary:**

This paper provides new generalization bounds for GCNs. The bounds are derived by evaluating the covering number of the hypothesis space in the Fourier domain of convolution operators. Specifically, when the spectrum is bounded, the bound does not depend on the number of trainable parameters. When the spectrum decays rapidly, the bound is independent of the number of nodes on a graphon, which can be interpreted as a graph with an infinite number of nodes.

**Strengths:**

1. (Originality) To the best of my knowledge, this paper is the first to derive a generalization performance bound for GCNs utilizing spectral decay.
2. (Quality) It provides a thorough review of previous statistical learning theory research on GCNs, clearly positioning this work within this line of research.
3. (Clarity) The writing is clear. The paper's structure is appropriate, and the mathematical descriptions are accurate. I had no difficulties in understanding the paper's main claims.
4. (Significance) For single-layer GCNs, the proposed method's performance bound achieves a better order with respect to node size than existing methods.

**Weaknesses:**

1. The derivation of bounds employs evaluation using the covering number, which is a relatively classical statistical learning theory method. Therefore, its novelty from this perspective is limited.
2. There is a discrepancy in the problem setting between the theoretical analysis and the numerical experiments. The theoretical analysis considers a problem setting where the graph signal and teacher signal are given in the I.I.D. setting. On the other hand, the numerical experiments consider a transductive node classification problem.

**Questions:**

I would like the authors to address the concerns raised in the Weaknesses section.

**Details Of Ethics Concerns:**

N.A.

---

> ### Author Response · Authors · 2025-11-21
>
> We thank the reviewer for their time and the constructive feedback. Below are our detailed responses to each of the weaknesses raised by the reviewer.
>
> ## Response to Weaknesses
>
> 1. In statistical learning theory the measure of complexity of a hypothesis space is crucial to derive generalization bounds. Using covering numbers to derive generalization bounds is merely a choice to measure the complexity of hypothesis space. We would like to clarify that our novelty lies in viewing the generalization of GCNs from the spectral perspective, which is fundamentally different from prior works that analyze generalization in the parameter space (refer to the papers cited in Table 1).
>
>    We emphasize that we make no claims of novelty by using covering numbers, but only an explanation for the leap obtained through spectral perspective (Line 200 - 208). We expect similar leaps can be obtained using other complexity measures such as Rademacher complexity if analyzed in the spectral domain. For example, Rademacher complexity has tight upper and lower bounds in terms of covering numbers by Dudley's entropy integral and Sudakov's lower bound, so one could expect similar style results to arise.
>
> 2. In transductive node classification each data point corresponds to a feature vector of a node and the target is the label of that node, which is a special case of our theoretical framework. In particular, we can view the inputs in transductive node classification as i.i.d. masked node features (as in Line 101 - 103) which still satisfies the i.i.d. assumption in our theoretical analysis. Therefore, we respectfully disagree with the reviewer that there is a discrepancy in the problem setting between theory and experiments.

---

> > ### Comment · Reviewer_8YBy · 2025-11-23
> >
> > Thank you for the clarification. It helps to better understand the positioning of your work.
> >
> > Regarding the Novelty: Thank you for clarifying the scope of your contribution. I understand that your main contribution is the spectral perspective on GCN generalization, and that the covering number argument was utilized as a standard measure of complexity.
> >
> > Regarding the Problem Setting:  I respectfully point out that in the transductive setting, the models are allowed to access the feature vectors of the test nodes during the training process (e.g., [1]). This is conceptually different from the standard i.i.d. setting, where the training and test processes are strictly separated.
> >
> > [1] Ran El-Yaniv, Dmitry Pechyony, Transductive Rademacher complexity and its applications, COLT07, https://dl.acm.org/doi/10.5555/1768841.1768858

---

> > > ### Author Response · Authors · 2025-11-25
> > >
> > > We thank the reviewer for reviewing our response and appreciate the provided reference.
> > > After reviewing the referenced paper, we would like to provide the following clarifications:
> > >
> > > - In our experiments, we use pre-defined train/test splits from the PyTorch Geometric library
> > > (https://pytorch-geometric.readthedocs.io/en/2.5.2/modules/datasets.html).
> > > During training, the model has access only to the training nodes and their features.
> > > The test node features are *not used* during training. This is a standard supervised learning
> > > setting for node classification tasks on graphs. Therefore, we believe it is appropriate
> > > to assume that the masking of nodes into train/test sets is done in an i.i.d. manner.
> > >
> > > - We acknowledge that the term "transductive" has been used incorrectly in our draft.
> > > In the referenced paper, transductive learning is defined as a setting where the model has
> > > access to the test data (features) during training. However, in our experiments, we *do not*
> > > use test node features during training. Therefore, our experimental setting is not transductive
> > > in the sense defined in the referenced paper. To avoid confusion, we have removed the term
> > > "transductive" in the updated version of our draft.
> > >
> > > With this clarification, we hope the reviewer is satisfied that there is no gap between theory
> > > and experiments. If needed, we are happy to provide the code for our experiments to verify
> > > these points and engage in further discussion.

---

### Official Review · Reviewer_LuGU · 2025-11-01

**Soundness:** 2
**Presentation:** 2
**Contribution:** 2
**Rating:** 4
**Confidence:** 4

**Summary:**

The paper investigates the generalization properties of Graph Convolutional Networks (GCNs) and attributes their empirical superiority over fully connected neural networks (FCNNs) to the spectral representation of graph convolution filters. The authors derive generalization bounds $O(\sqrt{\frac{n_x}{N}})$ that are independent of the number of parameters. They show that under mild spectral regularity conditions, GCNs can escape the curse of dimensionality. However, its immediate practical and theoretical claims are significantly weakened by several critical methodological flaws.

**Strengths:**

(1) The core idea that GCN generalization is determined by spectral complexity (covering numbers) rather than the parameter count is novel and addresses an open problem in GNN theory.

(2) If the results were fully valid, they would provide a rigorous theoretical explanation for a widely observed phenomenon: GCNs generalizing better than FCNNs on graph-structured data.

**Weaknesses:**

(1) Unrealistic Assumptions: The author assumes sub-Gaussian for graph signals and convex, smooth loss. In practice, graph node features are often sparse, categorical, or heavy-tailed. They are **not sub-Gaussian**. Classification losses (e.g., cross-entropy in **multi-class classification**) are **non-convex**. This limits the applicability of the theoretical bounds.

(2) Single-layer GCN analysis: All theoretical results in section 3.3 are derived for one layer $L=1$, whereas practical GCNs usually have 2 layers. **Multi-layer extensions are nontrivial**, and the bounds may degrade significantly due to Lipschitz composition and over-smoothing effects.

(3) Mismatch between theory and experiments: Experiments are conducted on ChebNet rather than standard GCNs, introducing a gap between the theoretical model and empirical validation. The **polynomial order $K$** in ChebNet affects receptive fields and generalization, which is not addressed.

(4) Negative generalization error in plots: Left one in Figure 4(a) shows **negative generalization errors** despite defining GE as absolute value in Eq. 4, suggesting either an inconsistency in the plot or a mismatch between theory and implementation.

**Questions:**

(1) How sensitive are your bounds to the assumption of sub-Gaussian node features, and can they be relaxed for sparse or categorical inputs?

(2) Can your $L=1$ layer analysis be generalized to multi-layer GCNs, and if so, how does the bound scale with $L$?

(3) Why were ChebNet models used in experiments instead of the theoretical GCNs, and how does the polynomial order $K$ affect the generalization?

(4) Explain the negative generalization error in left figure in Figure 4(a) when generalization error is defined as the absolute value in Eq. 4?

**Details Of Ethics Concerns:**

no.

---

> ### Author Response · Authors · 2025-11-21
>
> We thank the reviewer for their time and the constructive feedback. Below are our detailed responses to each of the weaknesses / questions raised by the reviewer.
>
> ## Response to Weaknesses / Questions
>
> 1. We respond to this concern in two folds:
>
>    a) **Sub-Gaussianity:** We respectfully disagree with the reviewer regarding the strictness of sub-Gaussian assumption. In addition to our global response on this topic we also provide additional details here. First, Hoeffding's lemma states that any bounded random variable is sub-Gaussian. As a result, categorical distributions are indeed sub-Gaussian. Similarly, sparse distributions can also be sub-Gaussian as long as the non-zero entries are drawn from a sub-Gaussian distribution. Moreover, datasets in our experiments have high sparsity levels either in node features or graph connectivity (see Table 2 for details).
>
>    Many real-world datasets indeed satisfy the sub-Gaussian assumption. For instance, all the datasets used in our experiments consist of node features that are either categorical or a review score or quadratic tails which are sub-Gaussian (see Table 2, Appendix A.2). As our results stand even for unbounded node features drawn from sub-Gaussian distributions, we believe that our theoretical framework is much more general and applicable to real-world datasets.
>
>    Extending our theoretical framework to heavy-tailed distributions would lead to slower rates than the current $\mathcal{O}(\sqrt{n_x/N})$ rate as it requires different concentration inequalities (as stated in Line 293 -- 295). Except for financial data (de Miranda Cardoso et. al., '21, due to time series nature we do not consider them here), we are unaware of real-world graph datasets with such tail properties. We believe that results that are applicable to such heavy-tailed frameworks are purely of theoretical interest and have limited practical relevance.
>
>    b) **Convexity:** We thank the reviewer for the comment, but we note that there appears to be a misunderstanding regarding what we require to be convex. Specifically, we would like to clarify that in Assumption 2 (Line 233 - 234) the we require convexity of loss $\ell(\mathbf{y}, \Phi(\mathbf{H}, \mathbf{x}))$ with to the model **outputs**, $\Phi(\mathbf{H}, \mathbf{x})$ but not necessarily in $\mathbf{H}$ -- making it a practical assumption (for example, satisfied by cross-entropy). We have included this in the updated version.
>
> 2. We thank the reviewer for this comment. We note that our core result is valid for both single and multi-layer GCNs, and in our original draft we focused our discussion on single layer models for simplicity. Nevertheless, in the updated version of the paper we have generalized the discussion to focus on multi-layer models. It is true that the Lipschitz constants compose multiplicatively across layers. However, as shown in Figure 1(b) the generalization error saturates after a certain value of Lipschitz constant, suggesting that in practice the bounds could be much tighter than the worst-case analysis.
>
>    Regarding over-smoothing, if the reviewer is referring to the over-smoothing phenomenon in deep GCNs (as in Rusch et. al., '23), where the node features become indistinguishable as the number of layers increases. We respectfully disagree with the reviewer that over-smoothing affects generalization. If all the filter's spectrum are low-pass, then the hypothesis complexity is lower leading to better generalization. However, over-smoothing may have bad performance on the training set to begin with -- this is an issue with expressivity of the GCN but not of the generalization of the model.

---

> ### Author Response · Authors · 2025-11-21
>
> 3. We wish to emphasize that our framework applies to any graph filter, regardless of how it is parameterized, in a polynomial vector space (Equation (1)). Therefore, there are many choices of possible parametrizations (or basis functions) to represent such filters. ChebNet (Deffard et. al' 16) is one such choice that uses Chebyshev polynomial basis to represent graph filters. Therefore, this choice complies with our theoretical framework and ensures that our experiments are consistent with our theoretical setting.
>
>    While there are other potential choices for parameterization, such as a monomial basis, we chose ChebNet in the experiments for the following reasons:
>
>    - **Best approximator**: Ideally we would like to use filtering operation (Equation (1)) in our experiments to match the theoretical model. However, finite order monomial filters are not the best approximation to polynomial space. Rather, it is shown in Geddes, 1978 that Chebyshev polynomial basis is the unique min-max optimal basis for polynomial space. Therefore, to best approximate the theoretical model we need to use ChebNet.
>
>    - **Computationally efficient**: Equation (1) requires $\mathcal{O}(n_x^2)$, while using Chebyshev polynomial basis we can implement it in $\mathcal{O}(n_f |\mathcal{E}|)$ time using recurrence relations, where $|\mathcal{E}|$ is the number of edges in the graph (Hammond et. al' 11, Deffard et. al' 16). This is crucial for our experiments as we are working with large graphs with sparse connectivity, see Table 2 for more details.
>
>    - **Lack of expressivity in Kipf and Welling'17**: This variant of GCN uses first-order polynomial approximation. Therefore, is not a faithful representation of the theoretical model in Equation (1). Furthermore, it shown in Xu et. al' 18 that such GCNs have limited expressivity. Therefore, to have a fair comparison with FCNNs we need to use higher-order polynomial filters such as ChebNet.
>
>    Finally, in Figure 1(a) we show that generalization error is invariant to the polynomial order $n_f$ as predicted by our Corollary 2.
>
> 4. We thank the reviewer for pointing out the subtle mistake. This is a plotting error, that is, we mistakenly plotted the signed generalization error instead of absolute generalization error -- the figure is corrected in the updated version.

---

### Author Response · Authors · 2025-11-21
**Global Response**

We thank all the reviewers and area chair for the smooth review process and constructive feedback. Generalization theory is a notoriously difficult problem and earliest known work dates back to 1960s by Vladimir Vapnik and Alexey Chervonenkis. Despite significant effort over 60 years, it remains an open problem to derive tight non-asymptotic generalization bounds for the deep neural networks. Our work makes significant progress in this direction for GCNs by leveraging the spectral properties of graphs and removing the parametric dependence in the bounds. We provide theoretical explanation for the empirical success of GCNs over FCNNs on graph-structured data, which is first of its kind result we are aware of. In particular, the reviewers acknowledged the elegance of our draft, novelty of our spectral perspective, and the tightness of our bounds; below are some representative quotes from the reviews:

## Summarized Strengths

1. "*The paper is generally well-written and clearly motivated.*" -- Reviewer ggKt
2. "*The writing is clear. The paper's structure is appropriate, and the mathematical descriptions are accurate. I had no difficulties in understanding the paper's main claims.*" -- Reviewer 8YBy
3. "*The spectral viewpoint provides an elegant and under-explored angle for bounding generalization in GCNs.*" -- Reviewer ggKt
4. "*...this paper is the first to derive a generalization performance bound for GCNs utilizing spectral decay.*" -- Reviewer 8YBy
5. "*If the results were fully valid, they would provide a rigorous theoretical explanation for a widely observed phenomenon: GCNs generalizing better than FCNNs on graph-structured data.*" -- Reviewer LuGU
6. "*Theoretical results are complemented by numerical simulations ... verifying the dependence of generalization on Lipschitz constants and spectral properties.*" -- Reviewer 8PWR

Our work is a significant departure from prior works that analyze generalization in the parameter space. Like in all theoretical works, assumptions are necessary to make the analysis tractable. Despite this, we ensured that our assumptions are mild and close to practice. We addressed all the concerns raised by the reviewers in our detailed individual responses. Nevertheless, we make a few global comments along these lines.

1. **Sub-Gaussian data assumptions**:
   We respectfully disagree with the reviewers that sub-Gaussian assumption is potentially restrictive for real-world graph data. We remind the reviewers that either of the following conditions is sufficient for sub-Gaussianity to hold true:
   - Bounded random variables (common assumption in the literature).
   - Tails of the distribution decay at least as fast as Gaussian.

   All the real-world datasets used in our experiments consist of node features that fall under one of the above categories (see Table 2, Appendix A.2). We believe that most real-world datasets satisfy these conditions as well, and, in fact, we are unaware of real-world graph datasets with heavy-tailed distributions, except perhaps financial data being an exception (de Miranda Cardoso et. al., '21) due to time series nature we do not consider them here. Unlike previous works, which often assume variables are bounded, our bounds are valid for unbounded random variables drawn from sub-Gaussian distributions. Therefore, our assumptions are a strict relaxation of common prior assumptions, which we believe makes our theoretical framework much more general and closer to practice.

2. **Motto of the experiments**:
   Our experiments are designed to validate the trends in Section 3.3. Our theoretical framework suggests that controlling the spectral complexity of GCN filters can lead to improved generalization; we have demonstrated this empirically for various real-world datasets that are both heterophilous and homophilous. Furthermore, in the updated version we have compared the empirical generalization error with our theoretical bounds and validated the trends of our bounds.

Based on the constructive feedback from the reviewers, we have improved our draft in the following ways:

1. In section 3.3, our submitted version only contained discussion of our results for single-layer GCNs for the sake of brevity. We note that this was largely for simplicity of presentation, as our main result is also valid for multi-layer GCNs as well. Neverthessless, following the feedback from reviewers, in the updated version, we have generalized this discussion to the case of multi-layer GCNs.
2. We have introduced a synthetic experiment to compare empirical generalization error with our theoretical bounds (Figure 3).
3. We have clarified the definitions of constants such as $K, K'', L_{\mathcal{X}}, L_{\mathcal{H}}$ in the updated version.
4. We have corrected all the typos and plotting issues pointed out by the reviewers.

---

> ### Author Response · Authors · 2025-11-21
>
> We conclude our global response by reiterating that our work makes significant advances over prior works in terms of novelty, tightness of bounds, and practical relevance. Our assumptions are mild, close to practice, and drastically relaxed compared to prior works. We believe we have addressed all the concerns raised by the reviewers in our detailed individual responses but are happy to engage in further discussion with the reviewers if additional concerns exist. Finally, we would like to thank the reviewers for their time and effort in reviewing our work and improving the quality of our manuscript.

---

### Meta-Review · Area_Chair_LUYX · 2026-01-11

**Summary:**

There was agreement among reviewers that the paper’s spectral perspective on GCN generalization is novel and potentially impactful. However,  there were a couple of concerns that led most reviewers to be marginally negative or borderline which are resolved to a large extent during rebuttal.  The main concern among reviewers was on the soundness and scope of assumptions, alignment between theory and experiments, and clarity/interpretability of the theoretical results. In particular, multiple reviewers questioned whether the sub-Gaussian data assumption and related regularity conditions are realistic for real-world graph data, and whether violations would materially weaken the conclusions. A second recurring concern was the initial focus on single-layer GCNs, given that practical models are typically multi-layer, raising questions about scalability of the bounds and the impact of Lipschitz constant composition and over-smoothing.  Also,while easy to fix, some reviewers had concerns about presentation, such as dense notation, insufficient intuition for constants, and lack of direct comparison between empirical errors and theoretical bounds.

**Reviewer Concerns:**

The assumptions such as bounded and categorical features are sub-Gaussian were clearly stated, justified, and aligned with experiments. The rebuttal and revision explicitly extended the discussion to multi-layer GCNs, clarified how Lipschitz constants compose, and argued that over-smoothing affects expressivity rather than generalization bounds. Definitions of constants, typos, and notation issues were improved, and a new synthetic experiment comparing empirical error to theoretical bounds helped contextualize tightness.

**Reviewer Scores:**

While reviewer LuGU's most concrete issues such as assumptions, multi-layer extension, ChebNet choice, and plotting errors were directly addressed, though some skepticism about assumptions may persist.  Reviewer 8PWR explicitly stated support for acceptance after the rebuttal, conditioned on incorporating results in subsequent version.

---

### Decision · Program_Chairs · 2026-01-26

Reject